



# The potential of OCO-2 data to reduce the uncertainties in $CO_2$ surface fluxes over Australia using a variational assimilation scheme

Yohanna Villalobos[1,2], Peter Rayner[1,2], Steven Thomas[1], and Jeremy Silver[1]

[1]School of Earth Sciences, University of Melbourne, Australia
[2]ARC Centre of Excellence for Climate Extremes, Sydney, Australia

**Correspondence:** Yohanna Villalobos (yvillalobos@student.unimelb.edu.au)

**Abstract.** This paper addresses the question of how much uncertainties in $CO_2$ fluxes over Australia can be reduced by assimilation of total-column carbon dioxide retrievals from the Orbiting Carbon Observatory$-2$ (OCO-2) satellite instrument. We apply a four-dimensional variational data assimilation system, based around the Community Multiscale Air Quality (CMAQ) transport-dispersion model. We ran a series of observing system simulation experiments to estimate posterior error statistics of optimized monthly mean $CO_2$ fluxes in Australia. Our assimilations were run with a horizontal grid resolution of 81 km using OCO-2 data for 2015. We found that on average, the total Australia flux uncertainty was reduced by up to 40% using only OCO-2 nadir measurements. Using both nadir and glint satellite measurements produces uncertainty reductions up to 80%, which represents 0.55 $PgC\,y^{-1}$ for the whole continent. Uncertainty reductions were found to be greatest in the more productive regions of Australia. The choice of the correlation structure in the prior error covariance was found to play a large role in distributing information from the observations. Overall the results suggest that flux inversions at this unusually fine scale will yield useful information on the Australian carbon cycle.

## 1 Introduction

The future of climate change depends mainly on the trajectory of green-house gas concentrations in the Earth's atmosphere, in particular carbon dioxide ($CO_2$) (Arora et al., 2013). Emissions from fossil fuel, land-use and land use-change have added more $CO_2$ to the atmosphere than can be readily absorbed by the ocean and biosphere (Myhre et al., 2013). Quantifying the terrestrial- and ocean-atmosphere carbon exchange is relevant for understanding the carbon cycle and climate since they play an important role by absorbing more than half of anthropogenic $CO_2$ emissions (Ciais et al., 2013). Despite important progress in quantifying all the components in the global $CO_2$ carbon budget, the amount of carbon uptake and release by land component remains poorly constrained by biosphere models. Currently, future predictions from most of the Dynamic Global Vegetation Models (DGVMs) are highly uncertain about the behaviour of the carbon cycle (Sitch et al., 2008). Even though DGVMs simulate a cumulative carbon uptake by 2099, the magnitude of the uptake varies considerably among them, especially at regional scale (Sitch et al., 2013, 2015). Reducing the regional-scale $CO_2$ flux uncertainties in these biogeochemical models (Canadell et al., 2010, 2011) is crucial to ascertain more accurate estimates of future climate projections (Friedlingstein et al., 2006; Huntingford et al., 2009; Friedlingstein et al., 2014). Inverse modelling of $CO_2$ fluxes (Ciais et al., 2010; Rayner et al., 2019)



can potentially help to constrain these uncertainties (Chevallier et al., 2010b) by directly using information from atmospheric $CO_2$ concentrations (Chevallier et al., 2005a, 2007; Baker et al., 2010).

Several studies over Europe (e.g. Broquet et al., 2011) and North America (e.g. Peters et al., 2007) have used ground-based $CO_2$ measurements to estimate $CO_2$ surface fluxes, which offer an accuracy of about 0.1-0.2 ppm. Despite their relatively

small measurement error, in-situ observations have some disadvantages, such as limited spatial representativeness. In-situ measurements are traditionally located at remote sites, distant from strong sources and sinks of $CO_2$. Finally, the existing in-situ network leaves much of the world unobserved (Ciais et al., 2013). For instance, the sparseness and spatial inhomogeneity of the atmospheric $CO_2$ monitoring system in the tropics and Southern Hemisphere restricts the potential of global atmospheric inversions to constrain regional fluxes in continents such as South America, Africa and Australia (Gurney et al., 2002; Peylin

et al., 2013).

Satellite-based retrievals of total-column $CO_2$ have the potential to address some of these shortcomings, since they have much higher spatial coverage compared with surface networks (Rayner and O'Brien, 2001; Ciais et al., 2014). During the last decade, satellite-derived estimates of the column-average $CO_2$ mole fraction have improved considerably. Before this period, satellite-based instruments had limited ability to constrain surface $CO_2$ fluxes, since their measurements were more sensitive

to $CO_2$ mixing ratios in the middle to upper troposphere and not in the lower troposphere where surface $CO_2$ fluxes have their greatest influence (Chevallier et al., 2005b).

The Scanning Imaging Absorption Spectrometer for Atmospheric Cartography (SCIAMACHY; Burrows et al., 1995; Buchwitz et al., 2015), which operated aboard ENVISAT during 2002-2012, was one of the first instruments with a more uniform sensitivity to $CO_2$ throughout the atmospheric column (including the boundary layer) compared to earliest satellite instruments

(e.g. Chédin, 2003; Crevoisier et al., 2009; Kulawik et al., 2010). Despite being sensitive to the lower vertical column of atmosphere, its large nadir surface footprint (30 km by 60 km) and the low single-sounding precision (2-5 ppm) restricted its ability to quantify in detail sources and sinks of $CO_2$ (e.g. Reuter et al., 2014). In contrast to SCIAMACHY, the Greenhouse Gases Observing Satellite (GOSAT, launched on January 23, 2009) was the first satellite created to measure $CO_2$ concentration with sufficient precision and resolution to study surface sources and sinks of $CO_2$ (Hamazaki et al., 2004; Yokota et al., 2009). Its

smaller footprint (10.5 km at nadir) and high scan rate (approximately 10,000 soundings per day) has provided considerably more information about regional carbon fluxes in previously unobserved regions (e.g. Parazoo et al., 2013).

The Orbiting Carbon Observatory-2 OCO-2 (launched on July 2, 2014) was also designed to be sensitive to $CO_2$ concentrations in the planetary boundary layer, with a even smaller nadir footprint (1.6 km × 2.2 km) and a higher precision than GOSAT (Eldering et al., 2017). A recent study Liang et al. (2017) found that GOSAT had a mean bias of -0.62 ppm and a

precision of 2.3 ppm over 2014-2016, while the bias and precision of OCO-2 were 0.27 ppm and 1.56 ppm, respectively; moreover, OCO-2 offers a denser spatial coverage compared to GOSAT, both in space and time.

Since 2013, several studies have used GOSAT retrievals to estimate $CO_2$ fluxes over the globe using inverse modelling (Basu et al., 2013; Chevallier et al., 2014; Deng et al., 2014; Maksyutov et al., 2013), while just a few have used OCO-2 data (Basu et al., 2018; Crowell et al., 2019). Most of these studies use global models with a relatively coarse spatial

and temporal resolution. For instance, the set of global three-dimensional models included in Basu et al. (2018) typically





have horizontal resolutions in latitude-longitude grid-cells between $1°$ up to $5°$. Coarse-resolution models capture large-scale transport processes but do not take full advantage of high-frequency information collected in the continental interior (Geels et al., 2004). Uncertainties related to the simulation of large-scale transport lead to poorly constrained flux estimates Chevallier et al. (2014). Several studies (e.g., Geels et al., 2004, 2006; Göckede et al., 2010; Broquet et al., 2011; Lauvaux et al., 2012)

indicate that errors in the simulation of large-scale atmospheric transport can be reduced if the transport model is run at sufficiently high resolution. Some of these studies (e.g., Broquet et al., 2011) performed a regional-scale variational inversion of the European biogenic $CO_2$ fluxes on a 50 km resolution. Finer resolution models have the potential to be more successful since they can offer a better representation of surface $CO_2$ fluxes and variability, as well as a better simulation of the processes driving high-frequency variability of transport (Schuh et al., 2010).

In this study, we present a regional-scale, four-dimensional variational flux inversion system to assimilate OCO-2 retrievals. The study area here is Australia, chosen for the following three reasons. First, the current estimate of Australian $CO_2$ fluxes is highly uncertain, mainly due to the uncertainties in the net primary productivity (NPP) simulated by biosphere models (Haverd et al., 2013b; Trudinger et al., 2016). In general, uncertainties in these NPP estimates are mainly driven by errors in model parameters (e.g., parameters associated with the leaf maximum carboxylation rate or the amount of chlorophyll content

in plants; Norton et al., 2018). Second, Australia has a sparse in-situ $CO_2$ monitoring network (four stations operating in our study year of 2015), so the broader coverage offered by satellite data may help to constrain fluxes. Third, Australia has reasonable coverage of OCO-2 measurements due to relatively low cloud, and the presence of three Total Carbon Column Observing Network sites in the region provides good calibration/validation for the OCO-2 data in the region.

This paper aims to assess the likely uncertainty reduction for $CO_2$ fluxes over Australia using a series of observing system

simulation experiments (OSSEs) and to test our four-dimensional flux inversion scheme. The structure of this paper is as follows. Section 2 describes the flux inversions system, the OSSEs and the datasets used. Section 3 presents the main results found for our ensemble of inversions, such as degree of freedom for signal, percentage of uncertainty flux reduction at grid-cell scale and uncertainty flux reduction aggregated by land cover type over Australia. Section 4 describes three sensitivity experiments to test the robustness of our inversion. In Section 5 we further evaluate our inversion by using real data; essentially

a consistency test, this is done by comparing the posterior $CO_2$ concentrations with OCO-2 data for March 2015. Sections 6 and 7 discuss the sensitivity experiments and summarise our findings.

## 2    Methods and Data

The methodology to perform our OSSEs follows Chevallier et al. (2007). This randomization approach is illustrated in Fig. 1 and follows four successive steps. First, we need to specify fluxes (see Section 2.4), boundary conditions and initial conditions

as inputs to the forward model (see Section 2.5). These inputs define the "true" field that we attempt to recover in the inversion. We run the Community Multiscale Air Quality (CMAQ) model forward with these inputs to generate a four-dimensional concentration field. We sample the concentration field with the OCO-2 observation operator to generate perfect observations (see Section 2.3). The perfect observations are perturbed following the observational error statistics to generate the "pseudo-





observations" used in the inversion. Second, we perturb the "true" fluxes according to the prior uncertainty to generate the prior fluxes. Third, we perform the Bayesian inversion (see Section 2.1), using the prior fluxes and pseudo-observations. Finally, we repeat the process of adding random noise to generate prior fluxes and pseudo-observations, and then running the flux inversion; these random realisations represent a sampling of the posterior error, taken as the difference between the posterior and true

fluxes. It can be shown that this difference is a realisation of a Gaussian distribution with zero mean and covariance given by the true posterior covariance.

In this study the OSSEs experiments were performed only for the months of March, June, September and December 2015. We ran an ensemble of five inversions for each month using different perturbations, generating five samples of the posterior PDF. In the following subsections we describe the main ingredients of this procedure.

## 2.1 Inversion Scheme

The inversion scheme for optimizing $CO_2$ surface fluxes over Australia involves a Bayesian four-dimensional variational assimilation system. The system is a generalised minimisation-based inverse-modelling framework, which can be applied to several potential models. We refer to it hereafter as 'py4dvar'. py4dvar finds an optimal estimate of the $CO_2$ surface fluxes ($x_a$) that fits both observations ($y$) and the prior fluxes ($x_b$) (Ciais et al., 2010; Rayner et al., 2019). Assuming Gaussian PDFs, finding

this maximum a posteriori estimate is equivalent to minimising the cost function $J(x)$ shown in Eq. 1 (Rayner et al., 2019).

$$J(\boldsymbol{x}) = \frac{1}{2}\left[(\boldsymbol{x}-\boldsymbol{x^b})^T \mathbf{B}^{-1}(\boldsymbol{x}-\boldsymbol{x^b})\right] + \frac{1}{2}\left[(\mathbf{H}(\boldsymbol{x})-\boldsymbol{y})^T \mathbf{R}^{-1}(\mathbf{H}(\boldsymbol{x})-\boldsymbol{y})\right] \tag{1}$$

The first term in Eq. 1 represents the sum of squared differences between the control variable ($x$) and its prior or background state ($x_b$). The second term measures the sum-of-squared difference between the model simulation, $\mathbf{H}(x)$, and observations ($y$) during the time window of the assimilation. The term $\mathbf{H}(x)$ is the function composition of an atmospheric transport

operator and an observation operator. Both terms in Eq. 1 are weighted by their respective error covariance matrices ($\mathbf{B}$ and $\mathbf{R}$), and the errors are assumed to be Gaussian and bias-free. As mentioned in the previous paragraph, the minimum of $J(x)$ is found by an iterative process rather than by an analytical expression. The minimization inside py4dvar is performed using the Limited-memory BFGS (L-BFGS-B) algorithm, as implemented in the `scipy` python module (Byrd et al., 1995). The minimization algorithm L-BFGS-B requires values of the cost function and its gradient, which are calculated using the CMAQ

forward model and the adjoint model, as shown in the third step in Fig. 1.

$$\nabla_x J = \mathbf{B}^{-1}(\boldsymbol{x}-\boldsymbol{x}^b) + \mathbf{H}^T(\mathbf{R}^{-1}\left[\mathbf{H}(\boldsymbol{x})-\boldsymbol{y}\right]) \tag{2}$$

The gradient of the cost function in Eq. 2 is calculated using the adjoint of the CMAQ model (version 4.5.1; Hakami et al., 2007). We can observe that in the second term in Eq. 2, the adjoint model ($\mathbf{H}(\mathbf{x})$) is applied to the vector $\mathbf{R}^{-1}(\mathbf{H}(\boldsymbol{x})-\boldsymbol{y})$, which is often called the "adjoint forcings", or simply the "forcings", and represents the error-weighted differences between

the forward model and the observed concentrations. Applying the adjoint model to the forcings, running backward in time from $t_{i-i}$ to $t_0$, allows us to construct the gradient of the cost function, $\nabla_x J(\boldsymbol{x})$.





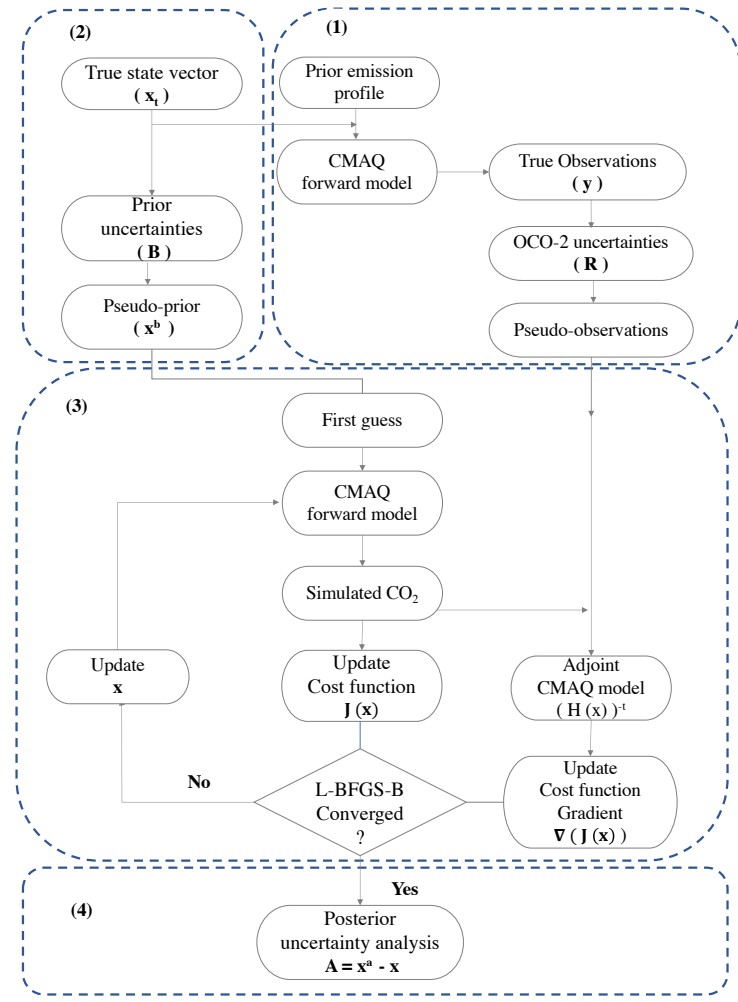

**Figure 1.** Diagram representing an overview of the Observing System Simulation Experiments (OSSEs) and how the inversion is performed using the L-BFGS-B minimisation algorithm.

## 2.2 Choice of Control variables

Our underlying physical variables are the monthly-averaged fluxes at the spatial resolution of CMAQ ($\approx 81$ km). We do not split fluxes by day and night, consistent with only using daytime satellite observations, which not subject to much influence by diurnal cycles in $CO_2$ fluxes (e.g., Deng et al., 2014; Houweling et al., 2015). Like most previous studies (e.g., Chevallier et al., 2007; Baker et al., 2010; Basu et al., 2013; Crowell et al., 2019) we use spatially correlated prior uncertainties to account for systematic errors in flux estimates. The variables exposed to the minimiser are not the fluxes themselves, but rather multipliers for the principal eigenvectors of $\mathbf{B}$. We truncate the eigen-spectrum at 99% of the total variance; doing this significantly reduces the size of the control vector (relative to if the control vector was comprised of the fluxes at each grid-





cell). This requires a different number of eigenvectors for different months (Table 1). The length of the control variables for our sensitivity experiments are defined in Table 5.

**Table 1.** Length of the control vectors ($\boldsymbol{x}$) for each of the simulation months.

| Months | Control variables ($\boldsymbol{x}$) |
|---------|---------------|
| 2015-03 | 811 |
| 2015-06 | 822 |
| 2015-09 | 745 |
| 2015-12 | 716 |

## 2.3 Observations and their Uncertainties

We used OCO-2 level 2 satellite data (Lite file version 9), the latest OCO-2 product distributed by the National Aeronautics and

5 Space Administration (NASA) (available for download from https://oco2.gesdisc.eosdis.nasa.gov/data/s4pa/OCO2_DATA/). We used the column-averaged dry air mole fraction of $CO_2$, referred to as $XCO_2$. We selected bias-corrected data, as described by Wunch et al. (2011). We only used nadir soundings over land that were flagged as good quality except in some of our sensitivity experiments (described in Section 4), in which we also included glint mode data. We computed a weighted average for all OCO-2 measurements using a two-step process similar to Crowell et al. (2019). The first step is to average all the

10 soundings into 1-second intervals and the second is to average these 1-second averages into the CMAQ vertical columns (81 km $\times$ 81 km) for each satellite pass, where the transit time over the CMAQ grid-cell is about 11 seconds. For the 1-second averaging process, the weighted averaging is defined in Eq. 3.

$$\hat{x}_{CO_2} = \frac{\sum_{i=1}^{n} w_i \times x_{CO_2,\, i}}{\sum_{i=1}^{n} w_i} \tag{3}$$

where $w_i = \frac{1}{\sigma_i^2}$ is the squared reciprocal of the OCO-2 uncertainties ($\sigma_i$). To get the uncertainties of these averaged soundings,

we considered 3 different forms of uncertainty calculation (similar to Crowell et al. (2019)). First if we assumed that all errors are entirely correlated in a 1-second span, we can define the uncertainties as shown in Eq. 4.

$$\sigma_s^2 = \frac{1}{N} \left[ \sum_{i=1}^{N} \sigma_i \right]^2 \tag{4}$$

However, and because the average shown in Eq. 4 is sometimes low, we also considered the standard deviation of the $XCO_2$ measurements (here referred to as the spread, or $\sigma_r$, of the OCO-2 measurements). In other words, if the spread ($\sigma_r$) of the



$XCO_2$ measurements were higher than the $XCO_2$ uncertainty ($\sigma_i$), we used the spread value as shown in Eq. 5. We did this because the spread in OCO-2 measurements may reflect real differences across the field within a 1-second timespan.

$$\sigma_r^2 = \frac{1}{N} \sum_{i=1}^{N} [\bar{x}_{CO_2} - x_{CO_2, i}]^2 \tag{5}$$

Third, we also considered a baseline uncertainty ($\sigma_b$), based on an error floor ($\epsilon$) over land and ocean, as shown in Eq. 6. We did this because sometimes we did not have enough OCO-2 soundings to compute a realistic spread. The values for our baseline uncertainties were taken to be 0.8 and 0.5 ppm over land and ocean, respectively. Finally, and after defining the uncertainties for the 1-second averages, we choose the maximum value between $\sigma_s$, $\sigma_r$ and $\sigma_b$.

$$\sigma_b^2 = \left[ \frac{\epsilon_{base}^2}{N} \right] \tag{6}$$

The second step was to take these 1-second averages and average them within the CMAQ vertical columns using Eq. 7.

$$\bar{x}_{CO_2} = \frac{\sum_{j=1}^{n} w_j \times \hat{x}_{CO_2}}{\sum_{j=1}^{n} w_j} \tag{7}$$

where $w_i = \frac{1}{\sigma_j^2}$ represents the squared reciprocal square of the uncertainties average in the 1-second span ($\sigma_j$) and $J$ is the number of those 1-second values. The average uncertainty over the CMAQ domain (Eq. 8) was similar to the procedure outlined for 1-second average in Eq. 4. However, we also added a term to represent the contribution of the model uncertainty ($\sigma_m$). We assumed that the model had a uncertainty of about be 0.5 ppm. The observational error covariance matrix $\mathbf{R}$ was assumed to be diagonal.

$$\bar{\sigma}^2 = \frac{1}{J} \left[ \sum_{j=1}^{J} \sigma_j \right]^2 \tag{8}$$

After averaging the OCO-2 sounding over the CMAQ domain, we generated a set of pseudo-observations as described in step 1 of Fig. 1. In this process, we run the CMAQ model forward. We start with an assumed set of CMAQ inputs, which includes fossil fuel emissions, fires, land and ocean fluxes (see Section 2.4 for a description of these fluxes). Our py4dvar system takes in a vector $x$ representing perturbations to the assumed emission profile, which is set to all be zeros in the "true case", and converts it into a format accessible to CMAQ model (e.g., copying the monthly average values into the hourly resolution CMAQ model is configured to run with). These perturbations to the emissions (zero values in the "true" case) are then added to the assumed emission profile for CMAQ before the model is run to produce a four-dimensional $CO_2$ concentration field, as is in step 2 of Fig. 1. Fourth, this modelled $CO_2$ concentration field is then transformed using the OCO-2 observation space. Once is transformed, we perturbed the "true observations" with Gaussian random noise to generate pseudo-observations as follows.

$$y' = y_{sim} + \mathbf{R}^{1/2} \cdot p \tag{9}$$



The first term of Eq. 9, $y_{sim}$, represents the OCO-2 simulated observations using the "true" fluxes. The second term of Eq. 9 $p$ is a vector with the same size as $y_{sim}$ and contains normally distributed random numbers with mean zero and variance one. Scaling $p$ by the square root of $\mathbf{R}$ ensures that the resulting realisation has the assumed error distribution.

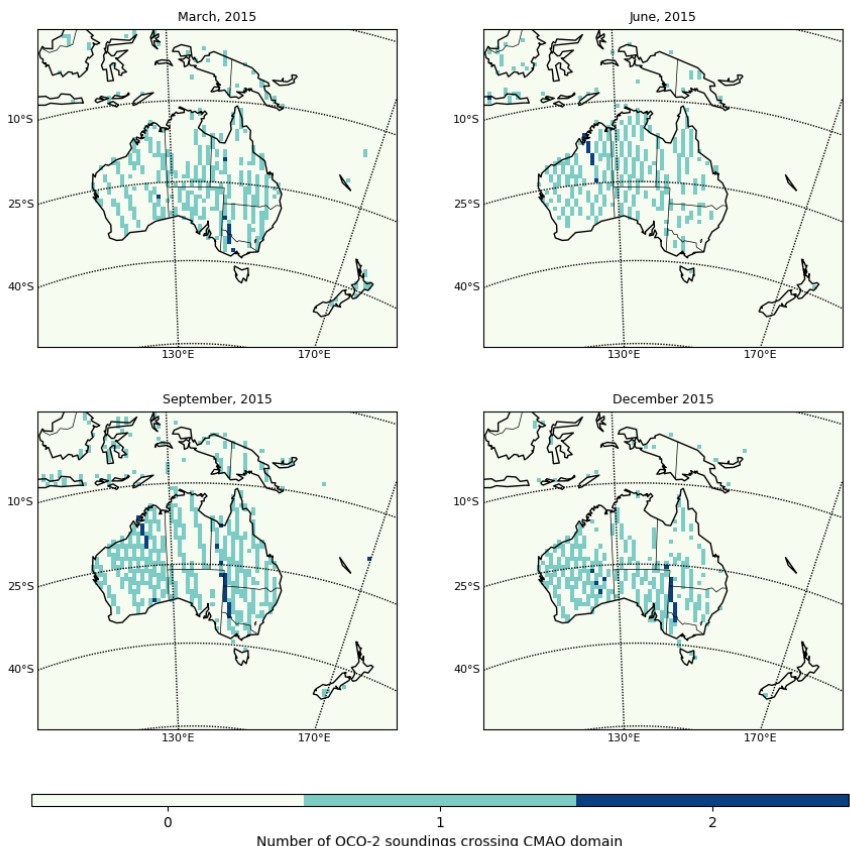

**Figure 2.** Spatial distribution of OCO-2 soundings over the CMAQ domain for March, June, September and December 2015.

## 2.4 Prior CO$_2$ fluxes and their uncertainties

5    As is stated in Section 2.5, the CMAQ model needs hourly emissions to run forward in time. We use the atmospheric convention that a negative flux value indicates an uptake by the surface and a positive value means a release of carbon to the atmosphere. Our total fluxes were comprised of four datasets representing elements of the CO$_2$ fluxes: terrestrial biospheric exchange, fossil-fuel, fires and air-sea exchange. Hourly biosphere CO$_2$ fluxes were calculated by combining two data sets: The Net





Ecosystem Exchange (NEE) at $0.5° \times 0.5°$ and daily resolution and the Gross Primary Production (GPP) at $0.5° \times 0.5°$ and 3-hourly resolution from the Community Atmosphere Biosphere Land Exchange (CABLE) model (Author, b).

The post-processing of 3-hourly NEE data involved four steps. First, we calculated daily GPP. Then we used daily GPP to estimate the daily Ecosystem Respiration (ER); in terms of carbon balance, the ER can be calculated as ER = GPP − NEE. Finally, daily ER was assumed equal throughout the day and subtracted from 3-hourly GPP to obtain 3-hourly NEE. These 3-hourly NEE fluxes were interpolated to hourly resolution. Recall that for our OSSEs, only the uncertainties, not the values themselves, are used. Given that the optimization was performed to optimize monthly fluxes, the uncertainties were computed with monthly resolution. We assumed that the biosphere flux uncertainties were equal to the Net Primary Production (NPP) simulated by CABLE, with a ceiling of $3$ gC m$^{-1}$ day$^{-1}$ following Chevallier et al. (2010a).

Fossil-fuel $CO_2$ emissions were obtained from the Fossil Fuel Data Assimilation System (FFDAS) (Rayner et al., 2010; Asefi-Najafabady et al., 2014). For this study, we used the 2015 FFDAS dataset (Author, a). The FFDAS uncertainty estimates were created by multiplying the FFDAS emissions dataset with a factor of 0.44. This factor was calculated by linear regression between the mean fluxes and the spread of an ensemble of 25 realizations of posterior $CO_2$ fluxes, following Asefi-Najafabady et al. (2014). We did not directly use those realizations to get the posterior FFDAS uncertainties, because the realizations only contained emissions over land (i.e., excluding domestic, aviation, and maritime emissions). These "missing" emissions were taken from the Emissions Database for Global Atmospheric Research (EDGAR) (Olivier et al., 2005). The highest value of FFDAS uncertainty over land was $2.3$ gC m$^{-2}$ day$^{-1}$ and over ocean $0.5$ gC m$^{-2}$ day$^{-1}$. This surprisingly large value over the ocean was a coastal point coinciding with Perth (Western Australia), where one of the largest and busiest general cargo ports in Australia is located.

Fire emissions were taken from the Global Fire Emission Database, version 4 (GFEDv4). This version of GFEDv4 provides gridded monthly fire emissions at $0.25°$ (van der Werf et al., 2017). The GFEDv4 product combines four satellite datasets: the Moderate Resolution Imaging Spectroradiometer (MODIS) burned area data product with active fires, data from the Tropical Rainfall Measuring Mission (TRMM) Visible and Infrared Scanner (VIRS) and the Along-Track Scanning Radiometer (ATSR). We used biomass-burning carbon emissions, a product based on GFEDv4 and the Carnegie Ames Stanford Approach (CASA) biosphere model (Randerson et al., 1996). Within the CASA model fire carbon losses are calculated for each grid cell and month, based on fire carbon emissions based on burned area from the GFED dataset. We assumed uncertainties for GFEDv4 corresponding to 20% of the biomass burning carbon emissions.

Ocean $CO_2$ fluxes were derived from the Copernicus Atmospheric Monitoring Service (CAMS) version 15r2 (Chevallier, 2016). The CAMS dataset is a global retrieval product, with a horizontal resolution of $3.75°$ in longitude and $1.875°$ in latitude at 3-hourly temporal resolution. Prior ocean fluxes estimated by CAMS were based on Takahashi et al. (2009). We assumed that the error statistics were uniform $0.2$ gC m$^{-2}$ day$^{-1}$ over ocean, as in Chevallier et al. (2010a).



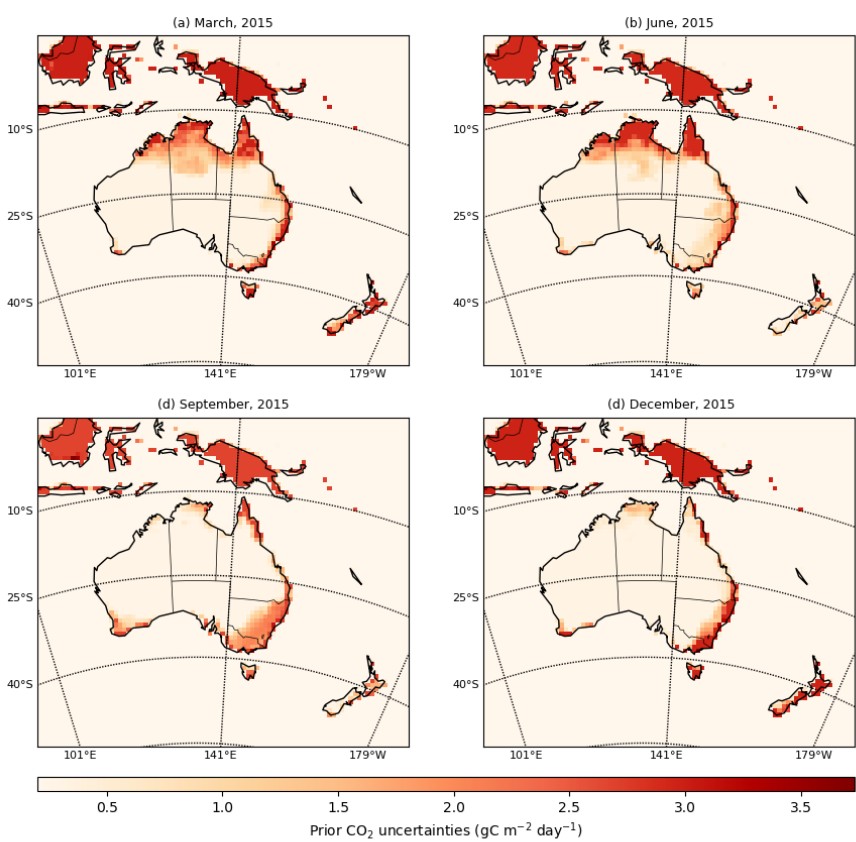

**Figure 3.** Monthly mean of $CO_2$ prior uncertainties accounting for the major terms in the $CO_2$ budget (anthropogenic fluxes, fires, land and ocean exchange), in units of $gC\ m^{-2}\ day^{-1}$.

After defining the emission profiles and their uncertainties, we incorporated spatial correlations into our prior error covariance matrix **B**. We assume no temporal correlations. This differs from Chevallier et al. (2010a) who used a temporal correlation length of four weeks, though this would only introduce weak correlations among our monthly-averaged fluxes. Following (Basu et al., 2013, section 3.1.1), the spatial correlation between grid-points $r_1$ and $r_2$ was defined as:

$$\mathbf{C}(r_1, r_2) = \exp^{-d(r_1,r_2)/\mathrm{L}} \tag{10}$$

where $d(r_1, r_2)$ is the distance (in km) between the two grid-points, and L, the correlation length, was assumed to be 500 km over land and 1000 km over ocean following Basu et al. (2013).





After defining $\mathbf{B}$, we performed an eigen-decomposition, $\mathbf{B} = \mathbf{W}^{\mathbf{T}}\mathbf{w}\mathbf{W}$, where $\mathbf{W}$ is a matrix of eigen-vectors and $\mathbf{w}$ is a diagonal matrix of corresponding eigenvalues. Figure 4a shows the cumulative percentage variance and demonstrates that 20 eigenvectors account for about $60\%$ of the variance in $\mathbf{B}$. We truncate the eigen-spectrum to retain 99% of the overall variance. The number required varied each month but was at most 400, compared to approximately 6,700 grid-points. The main reason

for this strong truncation is the large correlation length relative to the CMAQ grid resolution. We will test and discuss this later.

We solve the minimisation with a change of variable involving the eigen-vectors and normalising the by the square-root of the eigen-values; this transformation (given in Eq. 11) involves minisation with respect to $q$, rather than $x_p$. This step (often called pre-conditioning) accelerates convergence. It also simplifies the system since, all target variables have unit standard deviation. In our case, where we solve for perturbations around a background state, they also have a true value of zero. Generating our

prior flux for the inversion is achieved by defining a vector of normally distributed random numbers with unit standard deviation and zero mean. The process to generate the pseudo prior is represented in Eq. 11.

$$x_b = x_p + \mathbf{W}^{\mathbf{T}}\mathbf{w}^{1/2}q \qquad (11)$$

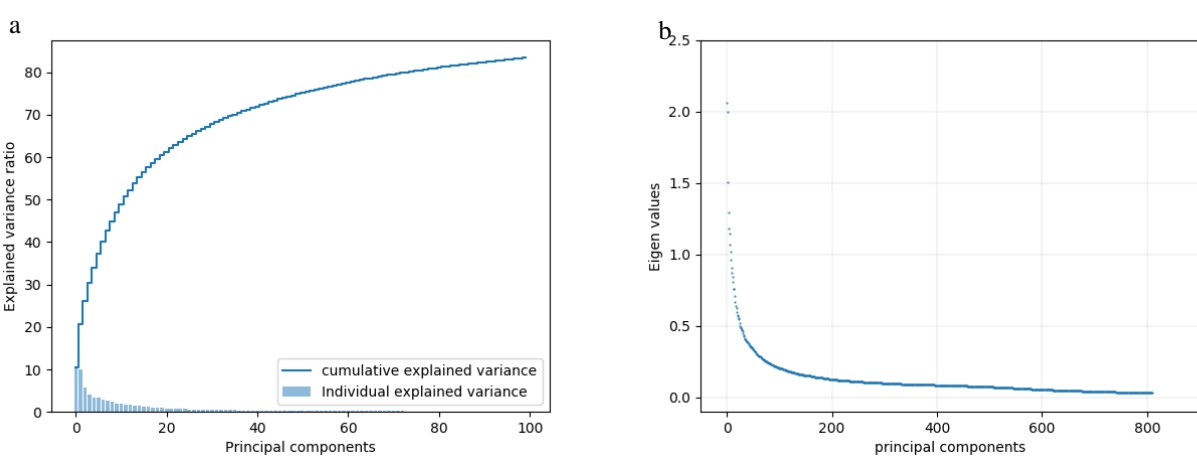

**Figure 4.** The cumulative percentage variance explained (left) and the eigenvalues (right) in the prior error covariance matrix.

## 2.5 CMAQ Model Configuration

We used the CMAQ modelling system and its adjoint (version 4.5.1; Hakami et al., 2007) to conduct numerical simulation of the

15 atmospheric $CO_2$ concentration over the Australian region. The CMAQ modelling system is an Eulerian (gridded) mesoscale Chemical Transport Model (CTM), initially created for air quality studies. It has been previously used to characterise the variability of $CO_2$ at fine spatial and temporal scales (Liu et al., 2014). The choice of an older version of the CMAQ modelling




system (cf. the latest version, v5.3) relates to the requirement of the model adjoint (needed to calculate the gradient of the cost function in the inversion).

We treat $CO_2$ as an inert tracer, neglecting its chemical production (Folberth et al., 2005; Suntharalingam et al., 2005). Thus modelled concentrations are determined only by emissions, the atmospheric transport (horizontal and vertical advection and diffusion), and initial and boundary conditions. Initial and boundary conditions were interpolated from atmospheric $CO_2$ concentration data from the Copernicus Atmospheric Monitoring Service (CAMS) global $CO_2$ atmospheric flux inversions Chevallier et al. (2010a). These data have a resolution of $3.75°$ in longitude and $1.875°$ in latitude with 39 vertical layers in the atmosphere; this dataset was also the basis for the oceanic fluxes used in the prior. The CMAQ chemical transport model (or CCTM) also requires 24-hourly three-dimensional emission data (recall that in our py4dvar system we solve for a perturbation around these background $CO_2$ fluxes). Here our background $CO_2$ fluxes were generated by adding the four $CO_2$ flux fields described in Section 2.4: carbon exchange between biosphere and atmosphere, carbon exchange between ocean and atmosphere, fossil-fuel emissions, and biomass burning emissions.

The CMAQ model is an off-line model, and thus requires three-dimensional meteorological fields as inputs for the transport calculations. We simulated meteorological data using the Weather Research and Forecast model (WRF) Advance Research Dynamical Core WRF-ARW (henceforth, WRF) version 3.7.1 (Skamarock et al., 2008). Details on the physics schemes used in our WRF configuration are shown in Table 2. Our domain has a horizontal resolution of 81 km and 32 vertical layers from the surface up to 50 hPa. The numerical simulation was carried out on a single domain (i.e., non-nested) of $89 \times 99$ grid-cells.

The meteorological initial conditions were based on the ERA-Interim global atmospheric reanalysis (Dee et al., 2011), which has a resolution of approximately 80 km on 60 vertical levels from the surface up to 0.1 hPa. Sea surface temperatures were obtained from the National Centers for Environmental Prediction/Marine Modeling and Analysis Branch (NCEP/MMAB). The WRF model was run with a spin-up period of 12 hours. The initial spin-up period stabilizes the model, that is, the inconsistencies between the initial and boundary conditions diminish in this period.

The WRF modelled meteorology was nudged towards the global analysis fields above the boundary layer. The default grid-nudging configuration was used; that is, nudging coefficients were assumed to be $10^{-4}\ \mathrm{s}^{-1}$ for wind and temperature and $10^{-5}\ \mathrm{s}^{-1}$ for moisture, as suggested by Deng and Stauffer (2006). Nudging has been widely used in mesoscale modelling as an effective and efficient method to reduce model errors (Stauffer and Seaman, 1990). It relaxes the model simulations of wind, temperature and moisture towards driving conditions, preventing model drift over a long-term integration.





**Table 2.** Physics parameterisations used in WRF model setup

| Category | Selected schemes |
| --- | --- |
| Microphysics | Morrison double-moment (Morrison et al., 2009) |
| Short wave radiation | Rapid Radiative Transfer Model (RRTMG) scheme (Iacono et al., 2008) |
| Long-wave radiation | Rapid Radiative Transfer Model (RRTMG) scheme (Iacono et al., 2008) |
| Surface layer | Monin-Obukhov (Monin and Obukhov, 1954) |
| Land/water surface | The NOAH land-surface model and the urban canopy model (Tewari et al., 2007) |
| Planetary Boundary Layercs (PBL) | Mellor–Yamada–Janjic scheme (Janjić, 1994)) |
| Cumulus | The Grell-Devenyi ensemble scheme (Grell and Dévényi, 2002) |

The WRF model output was post-processed by the Meteorology-Chemistry Interface Processor (MCIP) version 4.2 (Otte and Pleim, 2010). MCIP prepares the meteorological fields in a form required by CMAQ and performs horizontal and vertical coordinate transformation. In this process, we removed the outermost six rows and columns from each edge of the WRF model domain, so the horizontal CMAQ domain was set up (with $77 \times 87$ grid cells). This was done to prevent numerical instabilities in the "relaxation zone" (the exterior rows and columns of the horizontal domain), where the lateral meteorological boundary conditions and the WRF model's internal physical processes both contribute.

### 2.6 Observation Operator: CMAQ $CO_2$ simulations and OCO-2 measurements

As is seen in Eq. 1, we need to compare the CMAQ simulated $CO_2$ concentration with OCO-2 satellite retrievals. As outlined in Section 2.3, we averaged observations to approximate the observed XCO2 for any CMAQ grid-cell observed by OCO-2. To compare modelled and observed concentrations, we used the Eq. 12 (Rodgers and Connor, 2003; Connor et al., 2008)) to convolve the simulated $CO_2$ concentration with the relevant averaging kernels, as follows:

$$x^m_{CO_2} = x^a_{CO_2} - \sum_j \boldsymbol{h}_j \boldsymbol{a}_{CO_2,j} \boldsymbol{x}_a + \sum_j \boldsymbol{h}_j \boldsymbol{a}_{CO_2,j} \boldsymbol{x}^m_j, \tag{12}$$

where $x^a$ is the OCO-2 a priori, $\boldsymbol{h}$ is a vector of pressure weights, $\boldsymbol{h_j}$ is the mass of dry air in layer $j$ divided by the mass of dry air in the total column, $\boldsymbol{a}_{CO_2}$ is the averaging kernel of OCO-2, $\boldsymbol{x}_a$ is the OCO-2 a priori profile, and $\boldsymbol{x}^m$ is the simulated profile from the CMAQ model. In our py4dvar system, the first and second terms in Eq. 12 represent an "offset term". The OCO-2 averaging kernel is defined on 20 pressure levels and we interpolate these to the CMAQ vertical levels.

### 3 Results

In this section, we present an assessment of the uncertainty reduction resulting from the flux-inversion process. First, we present an analysis of the convergence of our minimization and evaluate the information content (degrees of freedom for signal) of



our OSSE simulation experiments. This is followed by an analysis of the uncertainty reduction categorized by MODIS land coverage. Finally, we present three sensitivity experiments to determine the robustness and consistency of our inversions.

### 3.1 Convergence Diagnostic

One interesting diagnostic of the convergence is how close the cost function comes to its expected theoretical value at the end
of the optimization. In a consistent system, the theoretical value of the cost function at its minimum should be close to half the number of assimilated observations, assuming all error statistics are correctly specified (Tarantola, 1987, p. 211). Table 3 shows the mean (across our five realisations) of the cost function and its gradient norm. With 420 observations, the theoretical value is 210, suggesting good convergence. The gradient norm decreased by 95%, suggesting some improvement is still possible. This percentage of reduction was found after iteration 10. We found little improvement on subsequent iterations. In a later
sensitivity experiment we will see that adding glint observations does indeed improve convergence.

**Table 3.** Convergence diagnostics of the inversion system using an ensemble of five independent OSSEs for March, June, September and December 2015.

| Months | Mean $J_0(\boldsymbol{x})$ | Mean $\nabla_x J_0$ | Mean $J_f(\boldsymbol{x})$ | Mean $\nabla_x J_f$ | % reduction $\nabla_x J$ | Mean DFS | N/2 |
|--------|------------|--------------|------------|-------------|------------------|----------|--------|
| 2015-03 | 299.58 | 897.65 | 219.95 | 47.34 | 94.73 | 21.54 | 210.00 |
| 2015-06 | 251.06 | 552.52 | 201.21 | 34.19 | 93.81 | 19.51 | 191.50 |
| 2015-09 | 298.08 | 580.16 | 244.08 | 35.03 | 93.96 | 24.71 | 246.00 |
| 2015-12 | 207.53 | 215.15 | 186.83 | 19.94 | 90.73 | 14.17 | 192.00 |

### 3.2 Degrees of Freedom for Signal

The number of degrees of freedom for signal (DFS) in our OSSEs is another useful diagnostic of the inversion (Rodgers, 2000, Eq. 2.46). The DFS quantifies the number of independent pieces of information that the OCO-2 measurements can provide given the prior information. In our experimental framework, we computed the DFS following (Chevallier et al., 2007, section
3.4.):

$$J(\boldsymbol{x}^a) = (\boldsymbol{x}^a - \boldsymbol{x}^b)^T \mathbf{B}^{-1} (\boldsymbol{x}^a - \boldsymbol{x}^b), \tag{13}$$

where $\boldsymbol{x}_a$ represents our posterior estimates. Table 3 shows that on average the DFS in the prior for our four months is about 20. This value is consistent with Fig. 4a and b, which shows that only about 20 eigenvalues account for 60% of the variance in our prior error covariance matrix. The inversion cannot add much information to other components, limiting the DFS. Australia
is a special case in this respect since most of the continent comprises semi-arid and arid regions. We assumed that land flux uncertainties are driven by NPP, as simulated by CABLE. Thus, the prior uncertainty will be small in arid and semi-arid regions.



### 3.3 Spatial distribution of uncertainty reduction

The uncertainty reduction between the posterior and prior fluxes is a useful way to evaluate the potential of satellite data to constrain $CO_2$ fluxes. We calculated the percentage uncertainty reduction following (Chevallier et al., 2007, section 3.5.), as follows:

$$\mathbf{U} = \left(1 - \frac{\sigma_a}{\sigma_b}\right) \times 100\% \tag{14}$$

where $\sigma_a$ and $\sigma_b$ are the posterior and prior standard deviations, respectively. Figure 5 displays the monthly uncertainty reduction in $CO_2$ fluxes for (a) March, (b) June, (c) September and (d) December 2015. We have masked areas with $\sigma_b < 10^{-7}$ mol m$^{-2}$ s$^{-2}$. We also mask areas with negative uncertainty reduction. Such uncertainty increase is simply a result of the small number of realisations. We will now describe the magnitude and spatial patterns in the uncertainty reduction, and in Section (3.4) we will discuss the uncertainty reduction aggregated by land cover class.

In March, the largest uncertainty reductions (Fig. 5a) are located in the north of Australia. In this area, the uncertainty reduction is greater than 30%, reaching values up to $60-70\%$. We note that the regions with the largest reduction in uncertainty coincide with the locations with high prior uncertainty (Fig. 3). In June 2015 (Fig. 5b), for instance, the largest uncertainty reduction was found in the north-west and south-east of Australia, where values range between $70-80\%$ and $60-70\%$ respectively. Uncertainty reduction in September (Fig. 5c) are higher compared to June in the Southern-East of the country. For instance, these values range between $70-80\%$. This is consistent with the fact that September is the in the middle of the growing season in this part of Australia and our prior uncertainties are driven by NPP. Also, more satellite soundings are available for this region in September compared to other months. The uncertainty reduction in December (Fig. 5d) decreases in the north of Australia to $20-30\%$. This is likely due to the fact that relatively few OCO-2 soundings are available in that month (Fig. 2), due to increased cloud coverage during the wet season in northern Australia. This is discussed further in the next section.



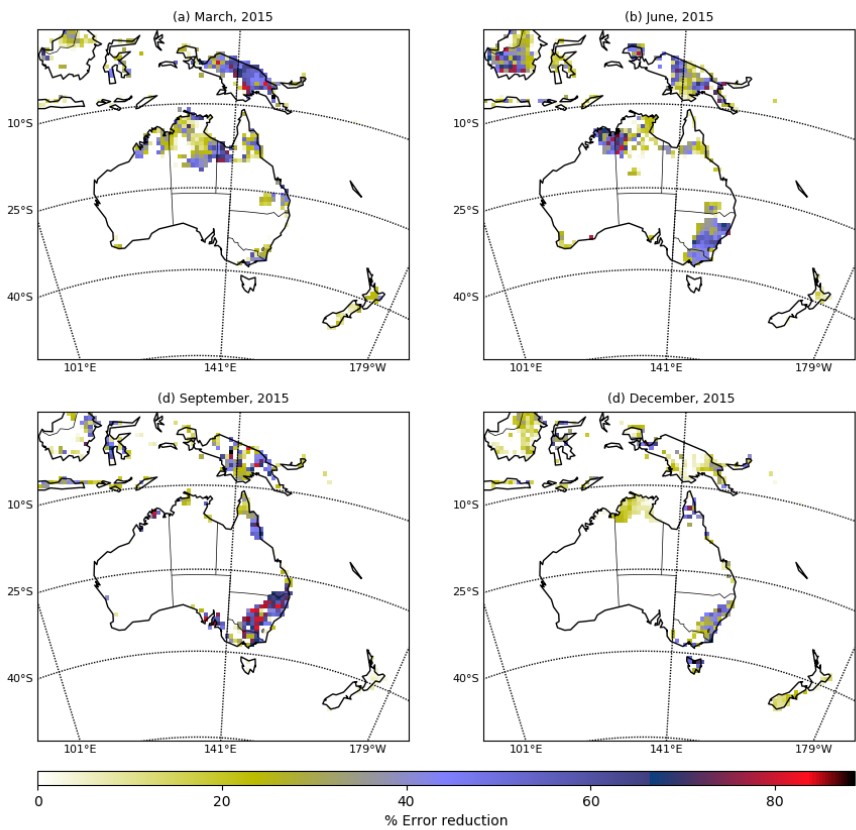

**Figure 5.** The percentage error reduction of the monthly mean $CO_2$ surface fluxes for March, June, September and December 2015 over the CMAQ model domain. The fractional error reduction is defined as $(1 - \sigma_a/\sigma_b)$, with $\sigma_a$ and $\sigma_b$ representing, respectively, the posterior and prior uncertainties of the $CO_2$ fluxes emissions.

### 3.4 Uncertainty reduction over Australia by MODIS land cover classification

To get a better understanding of the constraint on $CO_2$ surface fluxes provided by OCO-2, we aggregated the prior and posterior fluxes into six categories over Australia: grasses and cereal, shrubs, evergreen needle-leaf forest, savannah, evergreen broadleaf forest, and unvegetated land. We used the MODIS Land Cover Type Product (MCD12C1) Version 6 data product. The dis-
5   tribution is shown in Fig. 6. After aggregating fluxes for each realisation we calculated standard deviations and uncertainty reductions following Eq. 14.





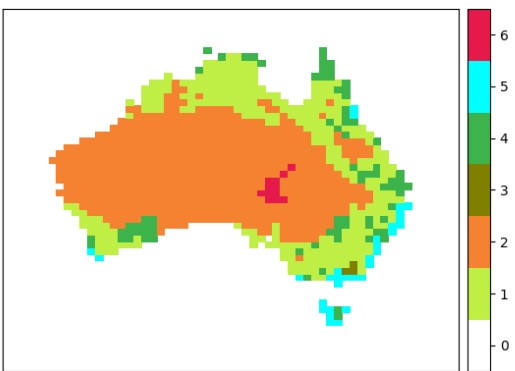

**Figure 6.** Aggregation of land cover classes over CMAQ domain using MODIS Land Cover Type Product (MCD12C1) Version 6 data product. Color bars represent each category: (0) ocean, (1) grasses and cereal, (2) shrubs, (3) evergreen needle-leaf forest, (4) savannah, (5) evergreen broadleaf forest, (6) unvegetated land.

The bar chart in Fig. 7 shows the prior and posterior flux uncertainties in $PgC\,y^{-1}$ along with the uncertainty reduction over Australia split into these five regions for (a) March, (b) June, (c) September, and (d) December 2015. The largest uncertainty reduction uncertainty in March is over grasses and cereals (72%), likely due to the relatively large NPP in that region (Fig. 3). Uncertainty reductions over savannah, evergreen broadleaf and evergreen needle-leaf forest are about 43%, 30% and 14%,

respectively. By contrast, we found no uncertainty reduction over shrubs and unvegetated areas. For this particular category, we found a negative error reduction; therefore, we set the posterior to be equal to the prior uncertainty. This unusual result is likely related to the small number of realizations performed. Also, Northern Australia has few soundings in March, probably due to cloudiness associated with the wet season.

June shows less uncertainty reduction for grasses and cereals (54%) likely due to the smaller number of OCO-2 soundings

(Fig. 2) in southern Australia. This region is also relatively cloudy in its winter season. By contrast, uncertainty reduction over the shrub ecotype increases, again following increased coverage. Even though relatively few soundings are found over evergreen broadleaf forest and evergreen needle-leaf forest in June, uncertainty reductions were 32% and 60%, respectively. The reduction over unvegetated areas is about 26%, again demonstrating the potential of OCO-2 data to constrain fluxes. For this month, we observe no uncertainty reduction over savannah, again for this category we set the posterior to be equal to prior

flux uncertainty.

The September OSSE was found to have higher prior uncertainties than all the other months, associated with the peak of the growing season in much of Australia. Uncertainty reductions are consequently larger, aided by increased OCO-2 coverage in south-eastern Australia. The uncertainty reduction over areas designated as savannah, evergreen broadleaf forest and evergreen



needle-leaf forest is about 61%, 64% and 39% respectively. Over areas classified as shrubs, we see a weaker uncertainty reduction of 48%.

The December OSSE yielded both smaller prior uncertainties and smaller uncertainty reductions. In this month areas classified as grasses and cereals showed an uncertainty reduction of about (40%). This is partly explained by fewer OCO-2 soundings being available in North and North-eastern Australia in that month. The scarcity of soundings in that area is likely due to cloudiness associated with the wet season (which generally spans November to April). Similar results are found over areas classified as savannah and evergreen broadleaf forest, where the uncertainty reductions were only 36% and 52%, respectively. Different results are seen over shrubs, where prior flux uncertainties are larger than the other months; the uncertainty reductions over this area are about (48%).

**Figure 7.** Prior and posterior uncertainties in $\mathrm{PgC\,y^{-1}}$ aggregated over five different classes over Australia domain using MODIS Land Cover Type Product (MCD12C1)





### 3.5 Uncertainty reduction in the total Australian CO$_2$ flux

Table 4 shows the standard deviation of the total CO$_2$ flux uncertainty over Australia for the four months in which inversions were run. We see reductions of 88% in September but only 40% in March. The differences are only partly explained by the combination of prior uncertainty and total number of soundings. For instance, the number of soundings in September is only

17% greater than in March. The soundings in September are denser over areas with high prior uncertainties such as grasses and cereals, savannah and evergreen broadleaf forest. These results suggest that the assimilation of OCO-2 retrievals can provide a significant constraint on estimates of Australia's carbon balance.

**Table 4.** Prior and posterior uncertainties in PgC y$^{-1}$ for an ensemble of five realizations aggregated over the Australia tontinent.

| Months | Prior (PgC y$^{-1}$) | Posterior (PgC y$^{-1}$) | Reduction % | Prior Reduction (PgC y$^{-1}$) |
|---|---|---|---|---|
| 2015-03 | 0.25 | 0.15 | 41 | 0.10 |
| 2015-06 | 0.44 | 0.18 | 59 | 0.26 |
| 2015-09 | 0.79 | 0.09 | 88 | 0.69 |
| 2015-12 | 0.63 | 0.29 | 54 | 0.34 |

## 4  Sensitivity Experiments

To assess the robustness and consistency of the previous results, we performed three different sensitivity experiments for March

2015. We analysed these using the same randomisation approach as our 'control case' (i.e., the OSSE presented above).

Sensitivity case 1 involved testing the effect of reducing the correlation lengths in our prior error covariance matrix. We changed the correlation length from 500 km to 50 km over land, and from 1000 km to 100 km over the ocean. By reducing the correlation length, the number of retained eigenvectors increased from 811 (control experiment) to 4101. The shorter correlation lengths allow a larger selection of possible flux structures, requiring more eigenvalues to capture the possible variance.

Sensitivity case 2 tested the effect of adding more observations to our inversion. Instead of using only nadir data ($\approx$ 420 soundings), we included glint observations over land and ocean ($\approx$ 1906 soundings). Here, the increase in the number of observations is about 365% on average.

In sensitivity case 3, we simplified the structure of **B**. We applied uniform uncertainties of 3 (PgC y$^{-1}$) over land and 0.2 (PgC y$^{-1}$) ocean and reduced the correlation length to 5 km over land and 10 km over ocean. This made **B** effectively

diagonal.



## 4.1 Degrees of Freedom for Signal

Table 5 shows the number of retained eigenvalues from **B** and the DFS for our three sensitivity experiments. Case 1 shows that merely reducing correlation lengths does not lead to extra information being resolved by the observations. Case 2 shows that, as expected, adding more observations resolves more information on fluxes. Case 3 (in which we reduce correlation lengths but also increase the uncertainty on many grid points) demonstrates an even greater increase in the number of components resolved by the observations. The comparison of cases 1 and 3 suggests it is the low uncertainty rather than the smoothness imposed by the uncertainty correlations that limits the DFS.

**Table 5.** Number of degrees of freedom for signal (DFS) in the prior flux uncertainty and the number the principal eigenvector in the prior error covariance matrix for three different OSSE sensitivity experiments.

| Sensitivity Experiments | Mean DFS | Principal Eigenvectors |
| --- | --- | --- |
| Control | 21.54 | 811 |
| Case (1) | 19.94 | 4101 |
| Case (2) | 39.08 | 811 |
| Case (3) | 53.04 | 3456 |

## 4.2 Spatial distribution of uncertainty reduction over Australia

Figure 8 shows the spatial distribution of the uncertainty reduction at grid-scale over Australia. These should be compared to Fig. 5a. Case 1 shown in Figure 8a indicates that the correlation length plays a significant role in the uncertainty reduction. A lower correlation length yields a lower reduction of the uncertainties. For example, the error reduction over the productive areas in northern and north-eastern Australia is between $(0-20\%)$ compared to the control experiment's $(40-80\%)$. This implies that longer correlation length-scales allow for information to be effectively "transferred" in space, thus pooling data over a wider region and magnifying the benefit from the assimilation.

Case 2 in Fig. 8b illustrates the benefit of adding more observations to the assimilation. The uncertainty reduction $(60-80\%)$ is much greater than the control experiment. These results complement Table 5, where the DFS increased from 21.0 (control experiment) to 39.1 (case 2).

Case 3 in Fig. 8c shows how the structure and magnitude of the prior uncertainty influence uncertainty reduction. The uncertainty reductions are distributed almost uniformly across Australia and their values range between $0-20\%$. Our assumption of a linear relationship between uncertainty and NPP means much of Australia has negligible impact on the uncertainty in the control case. This result shows the importance of that assumption. Assuming equal uncertainty across Australia may have a significant impact on the final total flux estimate in Australia, mainly because most the continent is largely composed of arid and semi-arid land. The small percentage of the uncertainty reduction is due to the negligible correlation length assumed in the prior error covariance matrix.





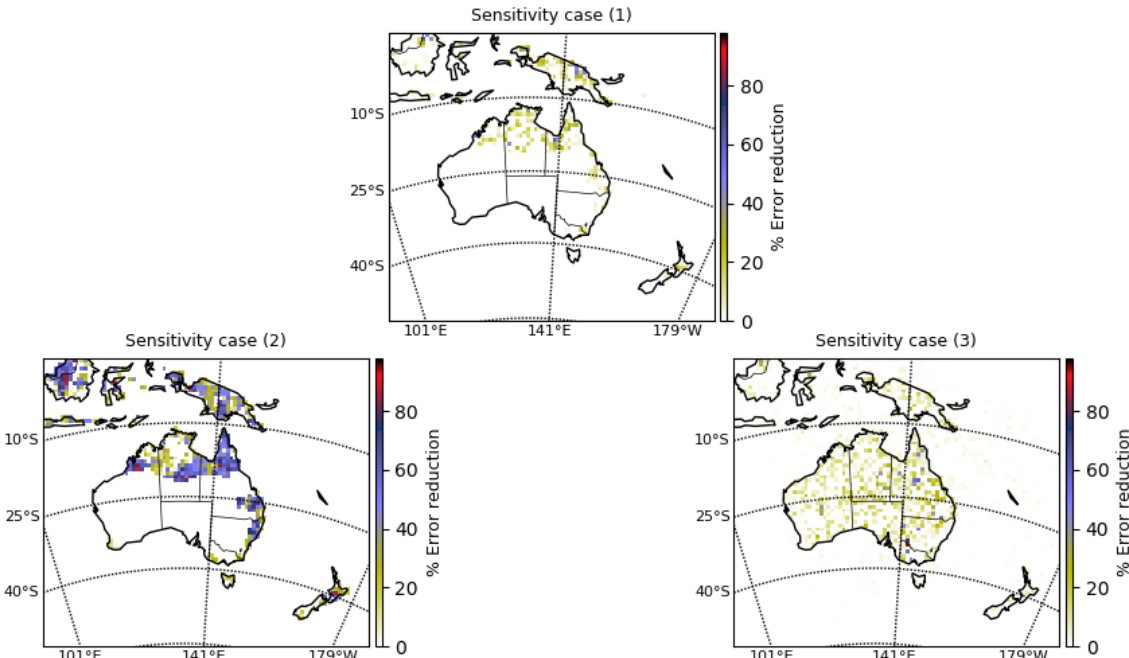

**Figure 8.** Maps of the percentage of error reduction for the three sensitivity cases. Top: using only nadir OCO-2 sounding and correlation lengths 50 km and 100 km. Left: using "nadir" and "glint" OCO-2 sounding and correlation lengths of 500 km and 1000 km. Right: uniform uncertainties over land and ocean, and correlation lengths 5 km and 10 km.

### 4.3 Uncertainty reduction over Australia by MODIS land cover classification

Fig. 9 shows the uncertainty reduction for the sensitivity cases aggregated by ecotype. There is good consistency between the geographical distribution (Fig. 8) and these spatial aggregates. Thus for case 1, the uncertainty reductions were found to be small compared to the results in the control experiment (Fig. 7a). For example, the sensitivity case 1 in Fig. 9a shows uncertainty

5   reductions over savannah and evergreen needle-leaf forest of about 2% and 16%, respectively. No uncertainty reductions are observed over shrubs and grasses and cereals.

Similarly, case 2 (Fig. 9b) displays significantly larger uncertainty reductions for the six land-use classifications compared to the control experiments ((Fig. 7a). For instance, the fractional uncertainty reductions over grasses and cereal reach values of about 74% and 80%, 58%, 35% over shrubs, savannah, and evergreen broadleaf forest, respecitvely. In the control experiment in

10   (Fig. 7a) these values only reach values of about 72%, 43% and 30% over grasses and cereal, savannah and evergreen broadleaf forest, respectively. As mentioned in the previous section, the stronger posterior reduction is due to the correlation length in





the prior covariance and an increase of the OCO-2 soundings over Australia. Findings in the sensitivity case 3 (Fig. 9c) shows similar results to those found in sensitivity case 1: the smaller the correlation length, the less efficient the inversion.

**Figure 9.** Sensitivity experiments for the prior and posterior uncertainties in PgC y$^{-1}$ aggregated over six different classes over Australia domain using MODIS Land Cover Type Product (MCD12C1)

## 4.4 Uncertainty reduction in the total Australia CO$_2$ flux uncertainty

Finally, we consider the uncertainty reduction of the total Australian CO$_2$ flux for our three sensitivity experiments. Results are presented in Table 6. Case 1 shows no uncertainty reduction compared to our prior fluxes. For this case, we set total posterior flux to be equal to prior. In this test, we can see again the importance of the choices of the correlation length in **B** before the optimization. We saw in Table 5 that by decreasing the spatial correlation to 5 km over land, we increase the number of





principal components. Given the small number of realizations and an increase in the number of components in the prior, we expect that this estimate of the uncertainty reduction may be less representative using our randomization approach.

Case 2 shows that by adding glint measurements and holding the correlation length of 500 km over land roughly doubles the control case's uncertainty reduction from 41% to 84%. This finding is significant for Australia, if such a system were used

to constrain the continent's $CO_2$ budget.

Case 3 demonstrates the same artefact as case 1, though the generally higher prior uncertainties in case 3 result in a higher uncertainty reduction for the total. Given this, the assimilation is still able to reduce the total uncertainty, to roughly the same value as case 1.

**Table 6.** Prior and posterior uncertainties in PgC y$^{-1}$ for an ensemble of five realizations

| Sensitivity cases | Prior (PgC y$^{-1}$) | Posterior (PgC y$^{-1}$) | Reduction % | Prior Reduction (PgC y$^{-1}$) |
|---|---|---|---|---|
| 1 | 0.14 | 0.14 | 0.0* | 0.00 |
| 2 | 0.66 | 0.12 | 83 | 0.55 |
| 3 | 0.20 | 0.13 | 32 | 0.06 |

Note: * indicates that the posterior uncertainty was set-up to be equal to prior uncertainty.

## 5    Comparison between CMAQ simulations and OCO-2 observations

One key uncertainty in any OSSE is the realism of the observational uncertainties. One simple test involves performing a limited inversion of data and assessing whether the cost function (Eq. 1) is consistent with the number of observations. Unlike the OSSE, this is not guaranteed; in the 'real-data' inversion, there are likely errors in the atmsopheric transport and the initial and boundary conditions. To test this, we performed an inversion for March 2015 using nadir data only. We added a scaling factor for the initial condition to our target variables for this test inversion. This avoids fluxes being unduly influenced by a

mismatch in initial concentrations. It is still consistent with the OSSE, since Peylin et al. (2005a) showed that the impact of the initial condition washed out of a domain the size of Australia in about five days and our real case inversion (the subject of a forthcoming paper) will cover at least one year.

Fig. 10 shows a histogram of residuals between the CMAQ model simulations using optimised fluxes and OCO-2 observations. We can see that the monthly mean bias was reduced from 0.50 to 0.01 ppm, with a decrease in the root mean square

error (RMSE) from 1.12 to 0.94 ppm. While these are based on the same data that were assimilated and do not necessarily show that the posterior fluxes are closer to the truth, it does show that our system is self-consistent. The cost function $J(\boldsymbol{x}^a)$ at its minimum is 219.95, close to half the number of observations (420).





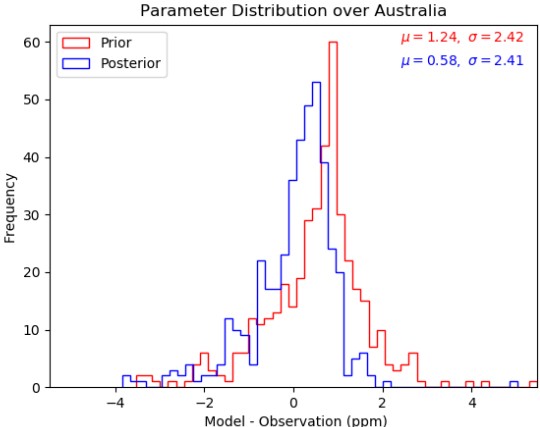

**Figure 10.** The distribution of the difference between simulated and observed $XCO_2$ in ppm. The red histogram presents the prior $XCO_2$ simulated minus the observed $XCO_2$, whereas the blue histogram presents the posterior $XCO_2$ simulated minus the observed $XCO_2$. Mean differences and standard deviations are indicated in the legend.

## 6  Discussion

In this paper, we quantified the potential uncertainty reduction in monthly $CO_2$ fluxes when assimilating OCO-2 satellite retrievals with a regional-scale model at approximately 80 km grid-resolution. If we compare our results shown in Fig. (5) against, for example, Figure 2 of Chevallier et al. (2007) we see that our grid-scale uncertainty reductions are higher than those

of Chevallier et al. (2007) by almost a factor of 2, using nadir data alone. Chevallier et al. (2007) demontrated uncertainty reductions of $30-50\%$ over productive regions in Australia while we see $60-80\%$. One possible explanation for this is the lower observational uncertainty assumed in our study, averaging 0.6 ppm compared with 2 ppm assumed by Chevallier et al. (2007) before OCO-2 was launched. We can also compare our results with those for the in-situ network studied by Ziehn et al. (2014). At the national scale, Ziehn et al. (2014) suggested an uncertainty reduction of 30% while we see 40% for our control

case.

Our results must be interpreted with caution because, like all OSSEs, they depend strongly on assumed inputs (such as **B** and **R**), which are difficult to characterize. In particular, we have assumed that the CABLE NPP (Haverd et al., 2013a) is a good proxy for biospheric net flux uncertainty, following Chevallier et al. (2010a). Chevallier et al. (2010a) used a different model and a different domain, so these assumptions may require further testing in our model configuration and region of interest.

In future, we could compare CABLE simulations against eddy-covariance $CO_2$ flux measurements following Chevallier et al. (2012). Characterization of the prior biospheric flux over semi-arid regions in Australia is critical to account for the inter-annual variability of these ecosystems (Poulter et al., 2014). Recent studies (e.g., Poulter et al., 2014) have suggested that the





semi-arid regions in Australia could become an important driver of the carbon cycle in comparison with ecosystems dominated by tropical rainforests.

Our sensitivity experiments (1) and (3) show that the uncertainty reduction in $CO_2$ surface fluxes over Australia is sensitive to a combination of both magnitude and spatial distribution of the uncertainty, as well as the choice of the correlation length-

scale. We saw in case (1), for example, that by reducing the correlation length in **B**, we do not necessarily increase the number of degrees of freedom (DFS) in our prior compared to the control. These findings suggest that the number of DFS in our prior fluxes depends more on the spatial distribution of error variance than on the assumed correlation length-scale. These results are much clearer in case (3), where the distribution of the uncertainty is uniform across Australia. In this case, we see that the number of DFS increases by increasing the magnitude of the uncertainty across Australia. In sensitivity case (2), we saw that

by including glint as well as nadir observations we significantly strengthen the prior flux constraint. Version nine of the OCO-2 data product shows no significant offset between nadir and glint observations, so future studies will use both measurement types (O'Dell et al., 2018).

Another important consideration in future work is such flux inversions should be run with a finer horizontal resolution. On the one hand, simulations with increased resolution have the potential to more accurately concentrations, thereby reducing the

model component of the observational uncertainty (Law et al., 2004; Peylin et al., 2005b; Patra et al., 2008). However, as we saw in Section 2.3, we found it necessary to average OCO-2 soundings before assimilating these data. To simplify this process, the averaging process removed any 1-second soundings that spanned multiple grid-cells in the CMAQ domain. This is about 7 km in along-track distance. If we use a finer resolution than 80 km, we could remove more soundings and thus weaken our constraint.

We emphasise again that our study quantifies the uncertainty but not the realism of our posterior flux estimates. The assessment of posterior fluxes from assimilation of real data will be the subject of an upcoming paper. This requires comparison with independent concentration data or, if available, flux estimates at comparable scales.

## 7  Conclusion

We have performed an observing system simulation experiment for the retrieval of $CO_2$ fluxes over Australia using OCO-

2 data and a regional-scale flux inversion system. The key findings were that OCO-2 nadir data can provide a significant constraint over the biologically active regions of Australia for most months. We saw that uncertainty reductions at grid-point scale over these productive areas can reach 90%. By contrast, there is not a significant reduction in uncertainties over arid and semi-arid regions, where the assumed prior uncertainties are small. For future work, it is relevant to consider a better characterization of our prior uncertainties in this region to account for the inter-annual variability of the carbon cycle in these

semi-arid regions. Sensitivity experiments show that uncertainty reductions are quite sensitive to the assumed prior correlations but less sensitive to the spatial distribution of prior uncertainties. These results also show that the glint data over land can add significant extra information. It seems likely, therefore, that this combination can help quantify the Australian carbon cycle, provided simulations are sufficiently realistic. Our future work will focus on the application of this assimilation system to





estimate $CO_2$ surface fluxes in Australia as a contribution to the Regional Carbon Cycle Assessment and Processes (RECCAP) project.

*Code availability.*   The py4dvar code was written by Steven Thomas and Peter Rayner and it can be found on GitHub. The code is available upon request from the authors.





## Appendix A: Convergence Diagnostic

**Table A1.** Convergence diagnostic of the inversion system using an ensemble of five independent OSSEs for March 2015 ($\nabla_x J_0$ and $\nabla_x J_0$ represents the initial cost function and its gradient at the beginning of the optimization, and $\nabla_x J_f$ and $\nabla_x J_f$ at the end of the optimization.

| | | | March, 2015 | | | | |
|---|---|---|---|---|---|---|---|
| Realizations | $J_0(x)$ | $\nabla_x J_0$ | N iterations | $J_f(x)$ | $\nabla_x J_f$ | % reduction $\nabla_x J$ | DFS |
| 1 | 413.99 | 1546.87 | 10 | 239.60 | 36.25 | 97.66 | 21.0 |
| 2 | 293.88 | 790.95 | 10 | 218.43 | 53.11 | 93.29 | 35.3 |
| 3 | 228.11 | 288.74 | 10 | 210.97 | 48.73 | 83.12 | 10.3 |
| 4 | 295.12 | 1042.63 | 10 | 215.55 | 34.00 | 96.74 | 18.0 |
| 5 | 266.78 | 819.06 | 10 | 215.20 | 64.58 | 92.11 | 23.2 |

**Table A2.** Convergence diagnostic of the inversion system using an ensemble of five independent OSSEs for June 2015 ($\nabla_x J_0$ and $\nabla_x J_0$ represents the initial cost function and its gradient at the beginning of the optimization, and $\nabla_x J_f$ and $\nabla_x J_f$ at the end of the optimization.

| | | | June, 2015 | | | | |
|---|---|---|---|---|---|---|---|
| Realizations | $J_0(x)$ | $\nabla_x J_0$ | N iterations | $J_f(x)$ | $\nabla_x J_f$ | % reduction $\nabla_x J$ | DFS |
| 1 | 247.70 | 522.25 | 10 | 195.85 | 26.49 | 94.93 | 23.07 |
| 2 | 234.26 | 367.03 | 8 | 194.85 | 32.31 | 91.20 | 17.80 |
| 3 | 208.09 | 232.12 | 10 | 182.55 | 27.50 | 88.15 | 17.67 |
| 4 | 329.57 | 1063.39 | 10 | 193.17 | 26.80 | 97.48 | 18.32 |
| 5 | 235.70 | 577.80 | 10 | 184.12 | 34.98 | 93.95 | 20.69 |

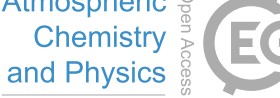



**Table A3.** Convergence diagnostic of the inversion system using an ensemble of five independent OSSEs for September 2015 ($\nabla_x J_0$ and $\nabla_x J_0$ represents the initial cost function and its gradient at the beginning of the optimization, and $\nabla_x J_f$ and $\nabla_x J_f$ at the end of the optimization.

| | | | September, 2015 | | | | |
|---|---|---|---|---|---|---|---|
| Realizations | $J_0(x)$ | $\nabla_x J_0$ | N iterations | $J_f(x)$ | $\nabla_x J_f$ | % reduction $\nabla_x J$ | DFS |
| 1 | 195.28 | 132.67 | 9 | 186.47 | 19.42 | 85.36 | 23.44 |
| 2 | 317.14 | 809.06 | 10 | 243.88 | 34.56 | 95.73 | 23.44 |
| 3 | 285.18 | 523.36 | 10 | 248.70 | 49.66 | 90.51 | 20.24 |
| 4 | 300.08 | 394.37 | 10 | 249.10 | 35.39 | 91.03 | 27.46 |
| 5 | 392.72 | 1041.33 | 10 | 292.27 | 36.12 | 96.53 | 28.96 |

**Table A4.** Convergence diagnostic of the inversion system using an ensemble of five independent OSSEs for December 2015 ($\nabla_x J_0$ and $\nabla_x J_0$ represents the initial cost function and its gradient at the beginning of the optimization, and $\nabla_x J_f$ and $\nabla_x J_f$ at the end of the optimization.

| | | | December , 2015 | | | | |
|---|---|---|---|---|---|---|---|
| Realizations | $J_0(x)$ | $\nabla_x J_0$ | N iterations | $J_f(x)$ | $\nabla_x J_f$ | % reduction $\nabla_x J$ | DFS |
| 1 | 182.79 | 156.16 | 8 | 167.15 | 19.70 | 87.39 | 19.21 |
| 2 | 249.54 | 419.86 | 8 | 200.60 | 18.09 | 95.69 | 16.73 |
| 3 | 196.66 | 107.93 | 10 | 190.33 | 22.37 | 79.27 | 10.68 |
| 4 | 194.24 | 231.99 | 9 | 177.95 | 19.36 | 91.66 | 10.36 |
| 5 | 214.43 | 159.79 | 8 | 198.13 | 20.18 | 87.37 | 13.89 |





## Appendix B: Uncertainty reduction over Australia classified by MODIS ecotype

**Table B1.** Uncertainty reduction of total $CO_2$ Australian flux in $PgC\,y^{-1}$ classified by MODIS ecotype (March, 2015).

| March, 2015 | | | | |
| --- | --- | --- | --- | --- |
| Land Cover type | Prior ($PgC\,y^{-1}$) | Posterior ($PgC\,y^{-1}$) | Reduction % | Prior Reduction ($PgC\,y^{-1}$) |
| Grasses/cereal | 0.176 | 0.049 | 72 | 0.127 |
| Shrubs | 0.045 | 0.045 | 0 | 0.000 |
| Savannah | 0.059 | 0.033 | 43 | 0.025 |
| Evergreen broadleaf forest | 0.078 | 0.055 | 30 | 0.023 |
| Evergreen needle-leaf forest | 0.014 | 0.012 | 14 | 0.002 |
| Unvegetated | 0.004 | 0.004 | 0 | 0.000 |

**Table B2.** Uncertainty reduction of total $CO_2$ Australian flux in $PgC\,y^{-1}$ classified by MODIS ecotype (June, 2015).

| June, 2015 | | | | |
| --- | --- | --- | --- | --- |
| Land Cover type | Prior ($PgC\,y^{-1}$) | Posterior ($PgC\,y^{-1}$) | Reduction % | Prior Reduction ($PgC\,y^{-1}$) |
| Grasses/cereal | 0.241 | 0.110 | 54 | 0.131 |
| Shrubs | 0.138 | 0.052 | 62 | 0.086 |
| Savannah | 0.060 | 0.060 | 0 | 0.000 |
| Evergreen broadleaf forest | 0.058 | 0.040 | 32 | 0.018 |
| Evergreen needle-leaf forest | 0.009 | 0.004 | 60 | 0.006 |
| Unvegetated | 0.003 | 0.002 | 26 | 0.001 |





**Table B3.** Uncertainty reduction of total $CO_2$ Australian flux in $PgC\ y^{-1}$ classified by MODIS ecotype (September, 2015).

| Land Cover type | September, 2015 | | | |
| --- | --- | --- | --- | --- |
| | Prior ($PgC\ y^{-1}$) | Posterior ($PgC\ y^{-1}$) | Reduction % | Prior Reduction ($PgC\ y^{-1}$) |
| Grasses/cereal | 0.378 | 0.078 | 79 | 0.300 |
| Shrubs | 0.095 | 0.049 | 48 | 0.046 |
| Savannah | 0.189 | 0.074 | 61 | 0.115 |
| Evergreen broadleaf forest | 0.160 | 0.058 | 64 | 0.102 |
| Evergreen needle-leaf forest | 0.010 | 0.006 | 39 | 0.004 |
| Unvegetated | 0.003 | 0.003 | 2 | 0.000 |

**Table B4.** Uncertainty reduction of total $CO_2$ Australian flux in $PgC\ y^{-1}$ classified by MODIS ecotype (December, 2015).

| Land Cover type | December, 2015 | | | |
| --- | --- | --- | --- | --- |
| | Prior ($PgC\ y^{-1}$) | Posterior ($PgC\ y^{-1}$) | Reduction % | Prior Reduction ($PgC\ y^{-1}$) |
| Grasses/cereal | 0.294 | 0.175 | 40 | 0.119 |
| Shrubs | 0.160 | 0.083 | 48 | 0.077 |
| Savannah | 0.094 | 0.060 | 36 | 0.034 |
| Evergreen broadleaf forest | 0.075 | 0.036 | 52 | 0.039 |
| Evergreen needle-leaf forest | 0.012 | 0.012 | 0 | 0.000 |
| Unvegetated | 0.004 | 0.003 | 6 | 0.000 |





## Appendix C: Sensitivity cases: Convergence Diagnostic

**Table C1.** Convergence diagnostic of sensitivity case (1) after the inversion using an ensemble of five independent OSSEs for March 2015 ($\nabla_x J_0$ and $\nabla_x J_0$ represents the initial cost function and its gradient at the beginning of the optimization, and $\nabla_x J_f$ and $\nabla_x J_f$ at the end of the optimization.

| March, 2015 | | | | | | |
|---|---|---|---|---|---|---|
| Realizations | $J_0(\boldsymbol{x})$ | $\nabla_x J_0$ | N iterations | $J_f(\boldsymbol{x})$ | $\nabla_x J_f$ | % reduction $\nabla_x J$ | DFS |
| 1 | 223.76 | 55.77 | 5 | 215.30 | 16.81 | 69.85 | 28.47 |
| 2 | 226.07 | 63.25 | 3 | 215.58 | 26.11 | 58.72 | 16.85 |
| 3 | 188.39 | 53.90 | 5 | 182.86 | 17.33 | 67.84 | 10.63 |
| 4 | 259.54 | 62.90 | 3 | 249.14 | 24.36 | 61.27 | 27.59 |
| 5 | 226.29 | 59.75 | 3 | 216.41 | 20.90 | 65.03 | 16.15 |

**Table C2.** Convergence diagnostic of sensitivity case (2) after the inversion using an ensemble of five independent OSSEs for Marc 2015 ($\nabla_x J_0$ and $\nabla_x J_0$ represents the initial cost function and its gradient at the beginning of the optimization, and $\nabla_x J_f$ and $\nabla_x J_f$ at the end of the optimization.

| March, 2015 | | | | | | |
|---|---|---|---|---|---|---|
| Realizations | $J_0(\boldsymbol{x})$ | $\nabla_x J_0$ | N iterations | $J_f(\boldsymbol{x})$ | $\nabla_x J_f$ | % reduction $\nabla_x J$ | DFS |
| 1 | 1896.70 | 6541.26 | 10 | 973.51 | 65.06 | 99.01 | 39.33 |
| 2 | 1355.84 | 3064.45 | 10 | 909.79 | 103.49 | 96.62 | 47.30 |
| 3 | 1189.52 | 2636.68 | 10 | 915.67 | 94.46 | 96.42 | 28.22 |
| 4 | 1589.50 | 5099.78 | 10 | 991.69 | 85.00 | 98.33 | 27.73 |
| 5 | 1148.68 | 903.66 | 10 | 949.96 | 70.36 | 92.21 | 52.84 |





**Table C3.** Convergence diagnostic of sensitivity case (3) after the inversion using an ensemble of five independent OSSEs for March 2015 ($\nabla_x J_0$ and $\nabla_x J_0$ represents the initial cost function and its gradient at the beginning of the optimization, and $\nabla_x J_f$ and $\nabla_x J_f$ at the end of the optimization.

| | | | March, 2015 | | | | |
|---|---|---|---|---|---|---|---|
| Realizations | $J_0(\boldsymbol{x})$ | $\nabla_x J_0$ | N iterations | $J_f(\boldsymbol{x})$ | $\nabla_x J_f$ | % reduction $\nabla_x J$ | DFS |
| 1 | 247.94 | 167.30 | 5 | 215.48 | 27.14 | 83.78 | 51.82 |
| 2 | 231.57 | 118.99 | 7 | 213.21 | 52.89 | 55.56 | 56.10 |
| 3 | 275.43 | 165.16 | 6 | 245.55 | 83.26 | 49.59 | 60.36 |
| 4 | 236.43 | 230.05 | 7 | 207.34 | 72.95 | 68.29 | 45.46 |
| 5 | 251.13 | 320.78 | 5 | 203.77 | 64.34 | 79.94 | 51.48 |





## Appendix D: Sensitivity cases: Uncertainty reduction of the total $CO_2$ Australian flux classified by MODIS ecotype

**Table D1.** Sensitivity Case (1): Uncertainty reduction of total $CO_2$ Australian flux in $PgC\,y^{-1}$ classified by MODIS ecotype (March, 2015).

| | March, 2015 | | | |
|---|---|---|---|---|
| Land Cover type | Prior ($PgC\,y^{-1}$) | Posterior ($PgC\,y^{-1}$) | Reduction % | Prior Reduction ($PgC\,y^{-1}$) |
| Grasses/cereal | 0.038 | 0.031 | 19 | 0.007 |
| Shrubs | 0.014 | 0.014 | 0 | 0.000 |
| Savannah | 0.025 | 0.024 | 4 | 0.001 |
| Evergreen broadleaf forest | 0.022 | 0.022 | 0 | 0.000 |
| Evergreen needle-leaf forest | 0.007 | 0.006 | 13 | 0.001 |
| Unvegetated | 0.001 | 0.001 | 2 | 0.000 |

**Table D2.** Sensitivity Case (2): Uncertainty reduction of total $CO_2$ Australian flux in $PgC\,y^{-1}$ classified by MODIS ecotype (March, 2015).

| | March, 2015 | | | |
|---|---|---|---|---|
| Land Cover type | Prior ($PgC\,y^{-1}$) | Posterior ($PgC\,y^{-1}$) | Reduction % | Prior Reduction ($PgC\,y^{-1}$) |
| Grasses/cereal | 0.228 | 0.062 | 73 | 0.166 |
| Shrubs | 0.219 | 0.026 | 88 | 0.192 |
| Savannah | 0.082 | 0.023 | 72 | 0.059 |
| Evergreen broadleaf forest | 0.047 | 0.024 | 49 | 0.023 |
| Evergreen needle-leaf forest | 0.005 | 0.004 | 22 | 0.001 |
| Unvegetated | 0.003 | 0.000 | 94 | 0.003 |





**Table D3.** Sensitivity Case (3): Uncertainty reduction of total $CO_2$ Australian flux in $PgC\ y^{-1}$ classified by MODIS ecotype (March, 2015).

| Land Cover type | March, 2015 | | | |
| --- | --- | --- | --- | --- |
| | Prior $(PgC\ y^{-1})$ | Posterior $(PgC\ y^{-1})$ | Reduction % | Prior Reduction $(PgC\ y^{-1})$ |
| Grasses/cereal | 0.051 | 0.051 | 0 | 0.000 |
| Shrubs | 0.072 | 0.071 | 2 | 0.001 |
| Savannah | 0.061 | 0.057 | 7 | 0.004 |
| Evergreen broadleaf forest | 0.010 | 0.010 | 0 | 0.000 |
| Evergreen needle-leaf forest | 0.007 | 0.005 | 26 | 0.002 |
| Unvegetated | 0.011 | 0.011 | 0 | 0.000 |



*Author contributions.* YV performed all the OSSEs experiments, including pre- and post-processing of data, and was responsible for developing the paper. ST was the principal developer of the py4dvar code with overall scientific guidance and additional analysis code from PR and JS. PR and JS also contributed to the writing of the manuscript.

*Competing interests.* The authors declare that they have no conflict of interest.

5  *Acknowledgements.* Yohanna Villalobos acknowledges the support of The National Commission for Scientific and Technological Research (CONICYT) Becas Chile. This research also was aided by the Australian Research Council (ARC) of the Centre of Excellence for Climate Extremes (CLEX) (CE17010002). This project was undertaken with the assistance of resources and services from the National Computational Infrastructure (NCI), which is supported by the Australian Government.



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
