# Peer review of "The potential of OCO-2 data to reduce the uncertainties in CO2 surface fluxes over Australia using a variational assimilation scheme"

_Atmospheric Chemistry and Physics, 2019_

## Referee Comment (RC1) · Anonymous Referee #1 · 16 Jan 2020

General comments:

Authors apply a regional grid-based inversion system built around CMAQ model and its adjoint to conduct OSSE simulations of the CO2 flux uncertainty reduction for Australia using actual OCO-2 retrievals.

The work has high methodological value as authors give sufficient detail on the design and operation of the inverse modeling system, so that is can become valuable learning material for those interested in using surface and satellite observation data in the regional inverse modeling studies with the variational optimization approach. Useful results include the impact of increasing prior flux uncertainties versus changing the

spatial correlation length for fluxes. The manuscript is well written and appears to be suitable for publications after technical corrections.

Detailed comments:

Page 2 Line 29 Authors wrote, "Liang et al. (2017) found that GOSAT had a mean bias of -0.62 . . .". Different GOSAT retrievals have their own biases, so it would be fair to give more detail, mentioning which product was used and the version number.

Page 19 Line 3 Sentence "The differences are only partly explained by the combination of prior uncertainty and total number of soundings." Authors may need to mention that due to prevailing winds, surface flux footprints for many OCO-2 soundings made over Australia lay over arid land thus contributing little to uncertainty reduction.

Page 25 Lines 15-18 Removing more observations on the edges of the grid cell in case of finer resolution does not seem to be the only possible way of mapping observations to the model grid. This limitation can be omitted from discussion.

Technical corrections

Page 7 Line 11 In a sentence which is related to Eq 7 it is written "J is the number of those 1-second values", while in the Eq. 7 the sum runs from 1 to n, so it is likely that n should be in place of J. On the contrary J appears as a number of elements in the next Eq. 8.

Page 7 Line 14 Omit "be" in "uncertainty of about be 0.5 ppm"

Page 10 Figure 3 caption: suggest writing as "prior $CO_2$ flux uncertainty" rather than "prior $CO_2$ uncertainty"

Page 16 Figure 5 caption: The statement on "The fractional error reduction is defined as . . ." looks somewhat out of place as figure shows percentage error reduction.

---

## Referee Comment (RC2) · Anonymous Referee #3 · 17 Jan 2020

This paper describes a regional flux inversion system to estimate fluxes over Australia with column CO2 observations from OCO-2. The authors test the performance and sensitivity of the system with a series of Observing system simulation experiments (OSSE). The performance of the system is primarily presented with the metric of uncertainty reduction assuming unbiased prior fluxes and pseudo observations. With increasing of satellite observations and the need to understand regional fluxes, the regional flux inversion is highly desirable. Therefore, the topic is important. The overall testing of the regional system roughly follows the traditional global inversion system, which I find is not sufficient. Though uncertainty reduction is a useful quantity to show the performance of the system, which highly depends on experimental setup as also

discussed in this paper. In the following, I suggest a few more experiments and other metric to test the sensitivity and performance of the regional inversions.

1. Different from global flux inversions, the regional flux inversions are sensitive to boundary conditions. I would suggest adding one experiment to show the sensitivity of the system to prescribed boundary conditions. For example, if the boundary conditions has random error of 1ppm, what does the result look like? Better yet is to assess the uncertainty of the boundary condition from CAMS, and then add that uncertainty in the OSSE.

2. Since the inversion assimilation window is short, the regional inversion must be sensitive to initial conditions as well. Therefore, testing the sensitivity of the system to initial condition and whether including the initial condition as part of state vector improve the performance would be very useful.

3. Satellites provide much denser observation coverage compared to surface CO2 observations, especially over tropics and the Southern Hemisphere. But at the same time, it is prone to bias in observations. The OSSEs are perfect to test the sensitivity of the inversion to potential bias in the observations. I suggest adding one experiment that assimilate biased pseudo observations. The bias could be based on the bias correction algorithm used in the OCO-2 retrieval products.

4. Unbiased prior fluxes certainly satisfy the theoretical assumptions in the variational optimization, but it is rarely the case in estimating land fluxes in atmospheric CO2 flux inversion. Scientifically, it is more useful to estimate a mean offset between the true fluxes and the prior fluxes. So I suggest to have a prior fluxes that have different mean values from the truth, and then test how the inversion could recover the mean fluxes.

Some minor comments:

1) I don't see the necessity to have section 5, since no real fluxes are presented. Also, the numbers on figure 10 are not consistent with the text.

2) The observation operator is different from several previous studies(e.g., Basu et al., 2013 cited in your paper). In equation (12), you interpret the averaging kernel to model levels. In a lot studies, the model vertical profiles are interpolated to the vertical levels of the retrievals, and pressure weighting function from retrievals is used in calculating model equivalent column $CO_2$. I think if the observation operator is done in this way, you will not have the problem having to remove 1-second averaging observations if they span several grids.

3) Line 6 on page 11, seems missing a word.

4) Line 3 on page 17, remove "uncertainty".

5) Line 3 on page 19, what could be other reasons? You used "partly" in the sentence.

6) Line 14 on page 25, double check the sentence. "the potential to more accurately observations"

---

## Referee Comment (RC3) · Anonymous Referee #2 · 9 Feb 2020

This manuscript is much improved over the previous submission. I think this is an important contribution, as it addresses many important questions about regional-scale inversions with satellite data, which to my knowledge has not been handled previously.

Other than a few minor revisions, I recommend publication.

Page 2, Line 15: "More uniform sensitivity" - More uniform than what? This is probably a reference to TES and AIRS, but need to be clear.

Page 6, line 4: Kiel et al (2019) is the best reference for the v9 data product

Page 9, Line 2: Missing reference "(Author, b)

[Figure]

Section 5: This is a bit unsatisfying, as the fluxes aren't reported. Is there a reason not to report the fluxes?

Page 25, Line 14: More accurately "simulate" concentrations?

---

## Author Comment (AC1) · 13 Feb 2020

**Responses to interactive comments on manuscript ACP-2019-874**

**Yohanna Villalobos Cortés**

**February 13, 2020**

This document presents a point-by-point reply to the reviewers comments on manuscript ACP-2019-874 (entitled 'The potential of OCO-2 data to reduce the uncertainties in $CO_2$ surface fluxes over Australia using a variational assimilation scheme'). Author Comment on behalf of all Co-Authors.

We would like to thank the reviewers for their comments and efforts towards improving our manuscript. The reviewer's comments are given in Roman type, and my replies are shown in blue.

**1  Response to referee #1**

**1.1  General comments**

Authors apply a regional grid-based inversion system built around CMAQ model and its adjoint to conduct OSSE simulations of the $CO_2$ flux uncertainty reduction for Australia using actual OCO-2 retrievals. The work has high methodological value as authors give sufficient detail on the design and operation of the inverse modeling system, so that is can become valuable learning material for those interested in using surface and satellite observation data in the regional inverse modeling studies with the variational optimization approach. Useful results include the impact of increasing prior flux uncertainties versus changing the spatial correlation length for fluxes. The manuscript is well written and appears to be suitable for publications after technical corrections

**1.2  Detailed comments**

1. Page 2 Line 29 Authors wrote, "Liang et al. (2017) found that GOSAT had a mean bias of -0.62". Different GOSAT retrievals have their own biases, so it would be fair to give more detail, mentioning which product was used and the version number.

   We have restructured the paragraph that start in line 29 on page 2

Initial text: A recent study Liang et al. (2017) found that GOSAT had a mean bias of -0.62 ppm and a precision of 2.3 ppm over 2014-2016, while the bias and precision of OCO-2 were 0.27 ppm and 1.56 ppm, respectively;moreover, OCO-2 offers a denser spatial coverage compared to GOSAT, both in space and time

Modified text: "A recent validation experiment, which compares GOSAT and OCO-2 against the Total Carbon Column Observing Network (TCCON) data (Liang et al., 2017) shows that in general OCO-2 has better accuracy in measuring the atmospheric $CO_2$ column concentration over 2014-2016. Liang et al. (2017) findings show that the mean biases of GOSAT (FTS Level 2-3 data products) were larger than OCO-2. Over 2014-2016, the GOSAT mean bias was of -0.62 ppm with a precision of 2.3 ppm compared to bias of OCO-2 (OCO-2 Lite File Product version 7), which was of 0.27 ppm with a precision 1.56 ppm. Because a wider detection coverage and higher spatial resolution, OCO-2 realize more accurate estimates of carbon dioxide. However, and despite these differences, both satellites on-orbit have atmospheric $CO_2$ detection capabilities to be used in regional atmospheric inversions to infer $CO_2$ surface fluxes"

2. Page 19 Line 3 Sentence "The differences are only partly explained by the combination of prior uncertainty and total number of soundings." Authors may need to mention that due to prevailing winds, surface flux footprints for many OCO-2 soundings made over Australia lay over arid land thus contributing little to uncertainty reduction

We have restructured the paragraph that start in line 3 on page 19

Initial text: The differences are only partly explained by the combination of prior uncertainty and total number of soundings. For instance, the number of soundings in September is only 17% greater than 5 in March. The soundings in September are denser over areas with high prior uncertainties such as grasses and cereals, savannah and evergreen broadleaf forest

Modified text: "The differences are only partly explained by the combination of prior uncertainty and total number of soundings. Another possible reason why we obtained a small percentage reduction in March compared to September is that in the northern region of Australia (the area where we assumed large uncertainties) winds come primarily from the north-west. Prevailing winds in this zone limit the ability of OCO-2 to constraint surface fluxes, mainly because we did not include OCO-2 soundings over the ocean. Second, the number of OCO-2 nadir soundings in September is 17% higher than March. Besides most of these soundings are located over land areas where we assumed that uncertainties were high such as grasses and cereals land, savannah and evergreen broadleaf forest.

3. Page 25 Lines 15-18 Removing more observations on the edges of the grid cell in case of finer resolution does not seem to be the only possible way of mapping observations

to the model grid. This limitation can be omitted from discussion.

**1.3 Technical corrections**

1. Page 7 Line 11 In a sentence which is related to Eq 7 it is written "J is the number of those 1-second values", while in the Eq. 7 the sum runs from 1 to n, so it is likely that n should be in place of J. On the contrary J appears as a number of elements in the next Eq. 8.

   Equation 7 has been corrected.

2. Page 7 Line 14 Omit "be" in "uncertainty of about be 0.5 ppm"

   Word "be" has been eliminated

3. Figure 3 caption: suggest writing as "prior CO2 flux uncertainty" rather than "prior CO2 uncertainty.

4. Page 16 Figure 5 caption: The statement on "The fractional error reduction is defined as..." looks somewhat out of place as figure shows percentage error reduction. In Figure 5 caption has been modified, we replaced "The fractional error reduction" with "The percentage of error reduction"

**2 Response to referee #2**

**2.1 General comments**

This manuscript is much improved over the previous submission. I think this is an important contribution, as it addresses many important questions about regional-scale inversions with satellite data, which to my knowledge has not been handled previously. Other than a few minor revisions, I recommend publication.

**2.2 Detailed comments**

1. Page 2, Line 15: "More uniform sensitivity" - More uniform than what? This is probably a reference to TES and AIRS, but need to be clear.

   We have restructured the paragraph that start in line 15 on page 2

   Initial text: The Scanning Imaging Absorption Spectrometer for Atmospheric Cartography (SCIAMACHY; Burrows et al., 1995; Buchwitz et al., 2015), which operated aboard ENVISAT during 2002-2012, was one of the first instruments with a more uniform sensitivity to $CO_2$ throughout the atmospheric column (including the boundary layer) compared to earliest satellite instruments (Chédin, 2003; Crevoisier et al., 2009; Kulawik et al., 2010)

Modified text: "The Scanning Imaging Absorption Spectrometer for Atmospheric Cartography (SCIAMACHY; Burrows et al., 1995; Buchwitz et al., 2015), which operated aboard ENVISAT during 2002-2012, was one of the first instruments with a more uniform sensitivity to $CO_2$ throughout the atmospheric column (including the boundary layer) compared to earliest satellite instruments such as the Operational Vertical Sounder (TOVS) (Chédin, 2003), the Infrared Atmospheric Sounding Interferometer (IASI) (Crevoisier et al., 2009) and the Tropospheric Emissions Spectrometer (TES) (Kulawik et al., 2010)"

2. Page 6, line 4: Kiel et al (2019) is the best reference for the v9 data product

   We have included the reference (Kiel et al., 2019)

3. Page 9, Line 2: Missing reference "(Author, b).

   We have updated the reference to (Harverd, 2018)

4. Section 5: This is a bit unsatisfying, as the fluxes aren't reported. Is there a reason not to report the fluxes?

   The assessment of posterior fluxes from assimilation of real data will be the subject of an upcoming paper.

5. Page 25, Line 14: More accurately "simulate" concentrations?

   We have restructured the paragraph that start in line 14 on page 19. We have added: "Another important consideration in future work is that these flux inversions should be run with a finer temporal and horizontal resolution. Model simulations at higher temporal and spatial resolution are always in better alignment with observation (fewer biases), mostly because they can sample closer to the measurement site location.

**3 Response to referee #3**

**3.1 General comments**

This paper describes a regional flux inversion system to estimate fluxes over Australia with column $CO_2$ observations from OCO-2. The authors test the performance and sensitivity of the system with a series of Observing system simulation experiments (OSSE). The performance of the system is primarily presented with the metric of uncertainty reduction assuming unbiased prior fluxes and pseudo observations. With increasing of satellite observations and the need to understand regional fluxes, the regional flux inversion is highly desirable. Therefore, the topic is important. The overall testing of the regional system roughly follows the traditional global inversion system, which I find is not sufficient. Though uncertainty reduction is a useful quantity to show the performance of the system, which highly depends on experimental setup as also discussed in this paper. In the following, I

suggest a few more experiments and other metric to test the sensitivity and performance of the regional inversions

**3.2 Detailed comments**

1. Different from global flux inversions, the regional flux inversions are sensitive to boundary conditions. I would suggest adding one experiment to show the sensitivity of the system to prescribed boundary conditions. For example, if the boundary conditions has random error of 1ppm, what does the result look like? Better yet is to assess the uncertainty of the boundary condition from CAMS, and then add that uncertainty in the OSSE.

   We agree that regional flux inversions are sensitive to lateral boundary conditions (BCs). The strength of the sensitivity depends on details such as domain size and distance from the boundaries to our region of interest. Ziehn et al. (2014, 2016), using a set-up rather like ours, found little sensitivity. We agree however that some sensitivity cases are warranted and are testing the impact of biased BCs on derived fluxes.

2. Since the inversion assimilation window is short, the regional inversion must be sensitive to initial conditions as well. Therefore, testing the sensitivity of the system to initial condition and whether including the initial condition as part of state vector improve the performance would be very useful

   This is probably not the case for two reasons. Firstly, Peylin et al. (2005) found little sensitivity for their European inversion to the initial condition. Sensitivity lasted about five days, comparable to the transit time for a concentration signal. Secondly, we are aiming to understand the uncertainty that will occur in real, much longer inversions. These last many months or even years. They are consequently not sensitive to initial conditions. However, we have decided to include this experiment in the paper (we will include the initial conditions as part of the state vector)

3. Satellites provide much denser observation coverage compared to surface $CO_2$ observations, especially over tropics and the Southern Hemisphere. But at the same time, it is prone to bias in observations. The OSSEs are perfect to test the sensitivity of the inversion to potential bias in the observations. I suggest adding one experiment that assimilate biased pseudo observations. The bias could be based on the bias correction algorithm used in the OCO-2 retrieval products.

   This is an excellent idea. We are running experiments using the differences between raw and bias-corrected $XCO_2$ as a bias term in our OSSE.

4. Unbiased prior fluxes certainly satisfy the theoretical assumptions in the variational optimization, but it is rarely the case in estimating land fluxes in atmospheric $CO_2$ flux inversion. Scientifically, it is more useful to estimate a mean offset between the true

fluxes and the prior fluxes. So I suggest to have a prior fluxes that have different mean values from the truth, and then test how the inversion could recover the mean fluxes.

This is also an excellent idea. We will run a biased prior case where we use the prior uncertainty not as a random perturbation (different for each realisation) but as a bias.

**3.3   Some minor comments**

1. I don't see the necessity to have section 5, since no real fluxes are presented. Also, the numbers on figure 10 are not consistent with the text.

   We disagree with this point. A common critique of OSSEs is that we have no way of assessing the input uncertainties. Comparing simulations and observations is one such check so we think it is important support to the results.

   Number in the figure 10 has been corrected.

2. The observation operator is different from several previous studies(e.g., Basu et al., 2013 cited in your paper). In equation (12), you interpret the averaging kernel to model levels. In a lot studies, the model vertical profiles are interpolated to the vertical levels of the retrievals, and pressure weighting function from retrievals is used in calculating model equivalent column $CO_2$. I think if the observation operator is done in this way, you will not have the problem having to remove 1-second averaging observations if they span several grids

   We agree that this approach would solve that problem. However CMAQ data is at higher vertical resolution than the 20 levels of the OCO2 retrieval. Running at least a simple interpolation from CMAQ to OCO-2 risks neglecting high resolution features in the CMAQ profiles. The averaging kernel is fairly smooth so the problem is less severe in this direction. It is a judgement call either way.

**3.4   Technical corrections**

1. Line 6 on page 11, seems missing a word.

   We have restructured the paragraph that start in line 6 on page 11. We have added: "We solve the minimization with a change of variable $\vec{x}^b$. Given that our control vector $\vec{x}$ depends on the size of the multipliers of the principal eigenvectors of $\mathbf{B}$. Our vector $\vec{x}^b$ was reconstructed (as is given in Eq.11). This reconstruction includes a new vector $\vec{q}$, which is normalized the by the square-root of the eigenvalues of $\mathbf{B}$; this transformation involves minimization with respect to $\vec{q}$, rather than $\vec{x_p}$."

2. Line 3 on page 17, remove "uncertainty".

   corrected

3. Line 3 on page 19, what could be other reasons? You used "partly" in the sentence..

This comment also was made by referee #1. We have restructured the paragraph that start in line 3 on page 19. We have added: "Another reason for a lower reduction in March compared to September is that in the northern region of Australia (the region where we assumed large uncertainties in March see Fig.3a) winds come from primarily from the west (active monsoon). Prevailing winds in this zone restrict the ability of OCO-2 to constraint surface fluxes (primarily because we did not include OCO-2 soundings over the ocean). Taking into account only the number of soundings in September, we can see that the increase of the OCO-2 data (17%) has a significant impact on the percentage of uncertainty reduction of the prior flux"

4. Line 14 on page 25, double check the sentence. "the potential to more accurately observations"

This comment also was made by referee 2. We have restructured the paragraph that start in line 14 on page 19. We have added: "Another important consideration in future work is that these flux inversions should be run with a finer temporal and horizontal resolution. Model simulations at higher temporal and spatial resolution are always in better alignment with observation (fewer biases), mostly because they can sample closer to the measurement site location".

**References**

Buchwitz, M., Reuter, M., Schneising, O., Boesch, H., Guerlet, S., Dils, B., Aben, I., Armante, R., Bergamaschi, P., Blumenstock, T., Bovensmann, H., Brunner, D., Buchmann, B., Burrows, J. P., Butz, A., Chédin, A., Chevallier, F., Crevoisier, C. D., Deutscher, N. M., Frankenberg, C., Hase, F., Hasekamp, O. P., Heymann, J., Kaminski, T., Laeng, A., Lichtenberg, G., De Mazière, M., Noël, S., Notholt, J., Orphal, J., Popp, C., Parker, R., Scholze, M., Sussmann, R., Stiller, G. P., Warneke, T., Zehner, C., Bril, A., Crisp, D., Griffith, D. W., Kuze, A., O'Dell, C., Oshchepkov, S., Sherlock, V., Suto, H., Wennberg, P., Wunch, D., Yokota, T., and Yoshida, Y. (2015). The Greenhouse Gas Climate Change Initiative (GHG-CCI): Comparison and quality assessment of near-surface-sensitive satellite-derived CO2 and CH4 global data sets. *Remote Sensing of Environment*, 162:344–362.

Burrows, J., Hölzle, E., Goede, A., Visser, H., and Fricke, W. (1995). Sciamachy—scanning imaging absorption spectrometer for atmospheric chartography. *Acta Astronautica*, 35(7):445–451.

Chédin, A. (2003). First global measurement of midtropospheric CO2 from NOAA polar satellites: Tropical zone. *Journal of Geophysical Research*, 108(D18):4581.

Crevoisier, C., Chédin, A., Matsueda, H., Machida, T., Armante, R., and Scott, N. A. (2009). First year of upper tropospheric integrated content of CO2 from IASI hyperspectral infrared observations. *Atmospheric Chemistry and Physics*, 9(14):4797–4810.

Harverd, V. (2018). personal communication.

Kiel, M., O'Dell, C. W., Fisher, B., Eldering, A., Nassar, R., MacDonald, C. G., and Wennberg, P. O. (2019). How bias correction goes wrong: measurement of xco2 affected by erroneous surface pressure estimates. *Atmospheric Measurement Techniques*, 12(4).

Kulawik, S. S., Jones, D. B., Nassar, R., Irion, F. W., Worden, J. R., Bowman, K. W., MacHida, T., Matsueda, H., Sawa, Y., Biraud, S. C., Fischer, M. L., and Jacobson, A. R. (2010). Characterization of tropospheric emission spectrometer (TES) CO2 for carbon cycle science. *Atmospheric Chemistry and Physics*, 10(12):5601–5623.

Liang, A., Gong, W., Han, G., and Xiang, C. (2017). Comparison of Satellite-Observed XCO2 from GOSAT, OCO-2, and Ground-Based TCCON. *Remote Sensing*, 9(10):1033.

Peylin, P., Bousquet, P., Le Quéré, C., Sitch, S., Friedlingstein, P., McKinley, G., Gruber, N., Rayner, P., and Ciais, P. (2005). Multiple constraints on regional CO2 flux variations over land and oceans. *Global Biogeochemical Cycles*, 19(1):1–21.

Ziehn, T., Law, R. M., Rayner, P. J., and Roff, G. (2016). Designing optimal greenhouse gas monitoring networks for Australia. *Geoscientific Instrumentation, Methods and Data Systems*, 5(1):1–15.

Ziehn, T., Nickless, A., Rayner, P., Scholes, R., and Engelbrecht, F. (2014). Greenhouse gas network design using backward Lagrangian particle dispersion modelling - Part 2: Sensitivity analyses and South African test case. *Atmospheric Chemistry and Physics*, 15(4):2051–2069.

---

## Author Comment (AC2) · 13 Feb 2020

We have attached a supplement PDF answering all the reviewers' comments (Please see supplement attached to referee #1 (acp-2019-874-AC1-supplement).

---

## Author Comment (AC3) · 13 Feb 2020

We have attached a supplement PDF answering all the reviewers' comments (Please see supplement attached to referee #1 (acp-2019-874-AC1-supplement)

---

## Author Response (AR1)

**Response to reviewers: ACP-2019-874**

**Yohanna Villalobos Cortés**

**March 24, 2020**

This document presents a point-by-point reply to the reviewers comments on manuscript ACP-2019-874 (entitled 'The potential of OCO-2 data to reduce the uncertainties in $CO_2$ surface fluxes over Australia using a variational assimilation scheme'). This reply is written on behalf of all Co-Authors.

We would like to thank the reviewers for their comments and efforts towards improving our manuscript. The reviewer's comments are given in Roman type, and my replies are shown in blue.

**1  Summary of changes**

The main changes in the manuscript can be summarised as follows:

1. We re-ran all OSSEs experiments using both land nadir and glint data version 9 (V9). In the first draft of the manuscript, the "the optimized fluxes" were estimated using only nadir land data (v9). Last year when we ran the OSSEs experiments we did not consider that "land nadir and glint observations can be treated as a single data set". Based on a personal communication by O'Dell (2019) that there were no systematic offsets between glint and nadir data we decided on including both data sets together. In this meeting, O'Dell showed that the new bias correction implemented in (v9) (Kiel et al., 2019) reduced significantly the offset between nadir and glint over land. Findings in Kiel et al. (2019) show the new bias correction in OCO-2's v9 reduces the standard deviation error over land from 1.35 ppm (version 8) to 0.74 ppm.

2. After re-running all the OSSE experiments, all figures and tables in the previous manuscript were updated. In addition to the green and orange bar in Figure 7, we added a purple bar which represents the prior uncertainties of 100 realizations (we did this to show how well our 5 realizations can represent the prior uncertainties). While we could only afford (computationally) a small number of the actual flux inversions, the prior realisations are quick to run and we could thus better sample this distribution.

3. We included the initial conditions (ICONs) in the control vector for all OSSEs experiments as recommended by reviewer #3. We added the following text to Section 2.2 (Choice of the control variables).

   Similar to Chevallier et al. (2005), and because our inversion assimilation window is short, we also include (in the state vector for the inversion) a perturbation to the initial conditions (ICONs) of the $CO_2$ concentration field. Because we are not interested in the analysis of this field, and in order not to significantly increase the size of the control vector, we added a scaling factor for the ICONs to our control variables . This scaling factor acts on the full three-dimensional concentration field. This avoids fluxes being unduly influenced by a mismatch in initial concentrations. We assumed 1% (approx. 4 ppm) uncertainties for these concentrations.

4. We expanded section 4 (sensitivity experiments). In this new version of the manuscript, we included 4 more sensitivity experiments as recommended by reviewer #3. All the names of the experiments were changed in the manuscript to S1, S2, S3, S4, S5, S6-A, S6-B. Section 4 starts with the description of the experimental design and changes made in the inversion as follows:

   **S1**: Test the effect of reducing the correlation lengths in our prior error covariance matrix **B**. The correlation length was changed from 500 km to 50 km over land, and from 1000 km to 100 km over the ocean. By reducing the correlation length, the number of retained eigenvectors increased from 811 (control experiment) to 4101. The shorter correlation lengths allow a larger selection of possible flux structures, requiring more eigenvalues to capture the possible variance.

   **S2**: Assess what percentage of uncertainty reduction of the Australian flux is affected by excluding glint land observations from our inversion. So far, all our OSSEs treat land nadir and glint data as one single dataset (because of the small offset between both ). The number of observations influences the footprint coverage, and therefore, the number of fluxes we can solve. In this particular experiment, we would expect a decrease in the error reduction over Australia because the number of observations has been reduced from 842 to 419 (50% on average).

   **S3**: Evaluate the effect of having uniform uncertainties over land and a simplified structure of **B**. In this case, we assumed uncertainties of 3 gC day$^{-1}$ over land with correlation of 5 km over land and 10 km over ocean. This transform **B** effectively in a diagonal matrix.

   **S4**: Test the impact of adding a mean absolute of 3.3 ppm bias to the OCO-2 observations. Here, biases were calculated by taking the differences between the raw and bias-corrected $XCO_2$ values found in OCO-2 retrieval product. We performed this experiment because some studies (e.g., Chevallier et al., 2007) indicate that just a few

tenths of a part per million bias in the observations are enough to prevent the inversions from converging on optimal fluxes.

**S5**: Test the impact of introducing a mean absolute bias of 0.21 PgC $y^{-1}$ to prior fluxes. In this experiment, the prior bias were created using a normal Gaussian random perturbation of the prior uncertainty. For all five realization, biases were introduced as constant component.

**S6-A**: Test the impact of adding bias in the boundary conditions (BCs). We increased the BCs simulated by adding a uniform offset of 0.5 ppm on each grid cell. In this case, we did not solve for BCs in the inversion.

**S6-B**: Assess the impact of incorporating BCs in the inversion system to deal with the bias introduced in S6-A. BCs were introduced to the control vector $\vec{x} = \{i_0, e_0, e_1, ..., e_n, b_0, ..., b7\}$ as eight boundary regions $b_0, ..., b7$ (representing the upper and lower areas of the North, South, East and West sides of the rectangular domain). We did not solved the BCs in the same way that we solve for the surface fluxes, as they are not among the key results (i.e., BCs were treated as nuisance variable). In this case, we gave the optimizer the ability to modify the BCs while it is optimizing surface fluxes. For this test, we assumed uniform uncertainty of $1.16e^{-5}$ ppm $s^{-1}$ (equivalent to 1 ppm $day^{-1}$). This is applied as an additive perturbation to temporally and spatially varying concentration boundary conditions based on the CAMS global $CO_2$ simulations.

5. Description of new sensitivity experiment results (S4, S5, S6-A and S6-B) are found in sections 4.5, 4.6, 4.7 and 4.8. In this section, we also included a table with a concise summary of these experiments.

6. We also included a supplementary document which show different wind roses for 10 different locations in the coastal area around Australia. Uncertainty reductions for coastal grid-points presents a problem for our inversion when the wind direction comes from the ocean (basically because our system only assimilates glint and nadir data over land).

7. We added more information to the discussion part (Section 7) which is related to the experiments where we included biases in the observation, BCs and prior fluxes.

**2 Response to referee #1**

**2.1 General comments**

Authors apply a regional grid-based inversion system built around CMAQ model and its adjoint to conduct OSSE simulations of the $CO_2$ flux uncertainty reduction for Australia using actual OCO-2 retrievals. The work has high methodological value as authors give sufficient detail on the design and operation of the inverse modeling system, so that is can become

valuable learning material for those interested in using surface and satellite observation data in the regional inverse modeling studies with the variational optimization approach. Useful results include the impact of increasing prior flux uncertainties versus changing the spatial correlation length for fluxes. The manuscript is well written and appears to be suitable for publications after technical corrections

**2.2   Detailed comments**

1. Page 2 Line 29 Authors wrote, "Liang et al. (2017) found that GOSAT had a mean bias of -0.62". Different GOSAT retrievals have their own biases, so it would be fair to give more detail, mentioning which product was used and the version number.

   We have restructured the paragraph that starts in line 29 on page 2

   Initial text: A recent study Liang et al. (2017) found that GOSAT had a mean bias of -0.62 ppm and a precision of 2.3 ppm over 2014-2016, while the bias and precision of OCO-2 were 0.27 ppm and 1.56 ppm, respectively; moreover, OCO-2 offers a denser spatial coverage compared to GOSAT, both in space and time

   Modified text: "A recent validation experiment, which compares GOSAT and OCO-2 against the Total Carbon Column Observing Network (TCCON) data (Liang et al., 2017) shows that in general OCO-2 has better accuracy in measuring the atmospheric $CO_2$ column concentration over 2014-2016. Liang et al. (2017) findings show that the mean biases of GOSAT (FTS Level 2-3 data products, version02.xx) were larger than OCO-2. Over 2014-2016, the GOSAT mean bias was -0.62 ppm with a precision of 2.3 ppm compared to OCO-2 biases (OCO-2 Lite File Product version 7), which was 0.27 ppm with a precision 1.56 ppm. Because a wider detection coverage and higher spatial resolution, OCO-2 realize more accurate estimates of carbon dioxide. However, and despite these differences, both satellites on-orbit have atmospheric $CO_2$ detection capabilities to be used in regional atmospheric inversions to infer $CO_2$ surface fluxes".

2. Page 19 Line 3 Sentence "The differences are only partly explained by the combination of prior uncertainty and total number of soundings." Authors may need to mention that due to prevailing winds, surface flux footprints for many OCO-2 soundings made over Australia lay over arid land thus contributing little to uncertainty reduction

   Given that we re-ran all OSSE experiments using land and glint data, we have restructured the whole paragraph that starts in line 3 on page 19

   Initial text: The differences are only partly explained by the combination of prior uncertainty and total number of soundings. For instance, the number of soundings in September is only 17% greater than in March. The soundings in September are denser over areas with high prior uncertainties such as grasses and cereals, savannah and evergreen broadleaf forest

Modified text: Table 7 shows the standard deviation of the total $CO_2$ flux uncertainty over Australia for the four months in which inversions were run. Months with the largest uncertainty reductions are found in December (80%), March (76%) and September (70%). In contrast with these results, the smallest reduction is found in June (31%). The last of these results is not surprising, since June is the month with smallest number of OCO-2 soundings (for this month we only find 694 observations compared to September and March, with 1002 and 842 soundings, respectively).

Differences in the uncertainty reduction between each month not only depend on the number of soundings and the structure of the uncertainty but also other variables (e.g. wind direction). Uncertainties for coastal grid-points presents a problem for our inversion when the wind direction comes from the ocean (basically because our system only assimilates glint and nadir data over land). Prevailing winds in this coastal zone restrict the ability of OCO-2 to constrain surfaces fluxes (Supplementary Figs. S1-S3).

3. Page 25 Lines 15-18 Removing more observations on the edges of the grid cell in case of finer resolution does not seem to be the only possible way of mapping observations to the model grid. This limitation can be omitted from discussion.

We agree that mapping observations to the model grid-cells are not the only approach. In general, global inversions often interpolate the model vertical profile to the vertical levels of the satellite retrievals and use the pressure weighting function from the retrieval to compute the modelled XCO2. We decided to do the interpolation in a different way because CMAQ vertical profile is at higher vertical resolution than the 20 levels of the OCO-2 retrieval. Running at least a simple interpolation from CMAQ to OCO-2 risks neglecting high resolution features in the CMAQ profiles. The averaging kernel is fairly smooth so the problem is less severe in this direction.

**2.3   Technical corrections**

1. Page 7 Line 11 In a sentence which is related to Eq 7 it is written "J is the number of those 1-second values", while in the Eq. 7 the sum runs from 1 to n, so it is likely that n should be in place of J. On the contrary J appears as a number of elements in the next Eq. 8.

Equation 7 has been corrected.

2. Page 7 Line 14 Omit "be" in "uncertainty of about be 0.5 ppm"

Word "be" has been eliminated

3. Figure 3 caption: suggest writing as "prior $CO_2$ flux uncertainty" rather than "prior $CO_2$ uncertainty.

We updated Figure 3 caption. We replaced "prior $CO_2$ uncertainty" with "prior $CO_2$ flux uncertainty"

4. Page 16 Figure 5 caption: The statement on "The fractional error reduction is defined as..." looks somewhat out of place as figure shows percentage error reduction.

In Figure 5 caption has been modified, we replaced "The fractional error reduction" with "The percentage of error reduction"

**3 Response to referee #2**

**3.1 General comments**

This manuscript is much improved over the previous submission. I think this is an important contribution, as it addresses many important questions about regional-scale inversions with satellite data, which to my knowledge has not been handled previously. Other than a few minor revisions, I recommend publication.

**3.2 Detailed comments**

1. Page 2, Line 15: "More uniform sensitivity" - More uniform than what? This is probably a reference to TES and AIRS, but need to be clear.

   We have restructured the paragraph that starts in line 15 on page 2

   Initial text: The Scanning Imaging Absorption Spectrometer for Atmospheric Cartography (SCIAMACHY; Burrows et al., 1995; Buchwitz et al., 2015), which operated aboard ENVISAT during 2002-2012, was one of the first instruments with a more uniform sensitivity to $CO_2$ throughout the atmospheric column (including the boundary layer) compared to earliest satellite instruments (Chédin, 2003; Crevoisier et al., 2009; Kulawik et al., 2010)

   Modified text: "The Scanning Imaging Absorption Spectrometer for Atmospheric Cartography (SCIAMACHY; Burrows et al., 1995; Buchwitz et al., 2015), which operated aboard ENVISAT during 2002-2012, was one of the first instruments with a more uniform sensitivity to $CO_2$ throughout the atmospheric column (including the boundary layer) compared to earliest satellite instruments such as the Operational Vertical Sounder (TOVS) (Chédin, 2003), the Infrared Atmospheric Sounding Interferometer (IASI) (Crevoisier et al., 2009) and the Tropospheric Emissions Spectrometer (TES) (Kulawik et al., 2010)"

2. Page 6, line 4: Kiel et al (2019) is the best reference for the v9 data product

   We have included the reference (Kiel et al., 2019)

3. Page 9, Line 2: Missing reference "(Author, b).

   We have updated the reference to (Harverd, 2018)

4. Section 5: This is a bit unsatisfying, as the fluxes aren't reported. Is there a reason not to report the fluxes?

   The assessment of posterior fluxes from assimilation of real data will be the subject of an upcoming paper.

5. Page 25, Line 14: More accurately "simulate" concentrations?

   We have restructured the paragraph that start in line 14 on page 19. We have added: "Another direction for future work would be to explore the impact of a finer temporal and horizontal on the resulting fluxes. Model simulations at higher spatio-temporal resolutions have been shown to have better agreement with observations, partly on account of allowing for a better representation of the measurements"

**4 Response to referee #3**

**4.1 General comments**

This paper describes a regional flux inversion system to estimate fluxes over Australia with column $CO_2$ observations from OCO-2. The authors test the performance and sensitivity of the system with a series of Observing system simulation experiments (OSSE). The performance of the system is primarily presented with the metric of uncertainty reduction assuming unbiased prior fluxes and pseudo observations. With increasing of satellite observations and the need to understand regional fluxes, the regional flux inversion is highly desirable. Therefore, the topic is important. The overall testing of the regional system roughly follows the traditional global inversion system, which I find is not sufficient. Though uncertainty reduction is a useful quantity to show the performance of the system, which highly depends on experimental setup as also discussed in this paper. In the following, I suggest a few more experiments and other metric to test the sensitivity and performance of the regional inversions

**4.2 Detailed comments**

1. Different from global flux inversions, the regional flux inversions are sensitive to boundary conditions. I would suggest adding one experiment to show the sensitivity of the system to prescribed boundary conditions. For example, if the boundary conditions has random error of 1 ppm, what does the result look like? Better yet is to assess the uncertainty of the boundary condition from CAMS, and then add that uncertainty in the OSSE.

   This experiment was included in manuscript. The divided this experiment into two experiments (S6-A and S6-B). For details please see section 4 (sensitivity experiments)

2. Since the inversion assimilation window is short, the regional inversion must be sensitive to initial conditions as well. Therefore, testing the sensitivity of the system to initial

condition and whether including the initial condition as part of state vector improve the performance would be very useful

*As mentioned at the beginning of this document, we decided to include the ICONs in all the OSSE experiments, and not only as another sensitivity experiment. Details of how we include this variable to the control vector is found in Section 2.2 (Choice of the control variables).*

3. Satellites provide much denser observation coverage compared to surface $CO_2$ observations, especially over tropics and the Southern Hemisphere. But at the same time, it is prone to bias in observations. The OSSEs are perfect to test the sensitivity of the inversion to potential bias in the observations. I suggest adding one experiment that assimilate biased pseudo observations. The bias could be based on the bias correction algorithm used in the OCO-2 retrieval products.

   *This experiment was included in manuscript. We defined this as S4. For details please see section 4 (sensitivity experiments).*

4. Unbiased prior fluxes certainly satisfy the theoretical assumptions in the variational optimization, but it is rarely the case in estimating land fluxes in atmospheric $CO_2$ flux inversion. Scientifically, it is more useful to estimate a mean offset between the true fluxes and the prior fluxes. So I suggest to have a prior fluxes that have different mean values from the truth, and then test how the inversion could recover the mean fluxes.

   *This experiment was included in manuscript. We defined this as S5. For details please see section 4 (sensitivity experiments).*

**4.3   Some minor comments**

1. I don't see the necessity to have section 5, since no real fluxes are presented. Also, the numbers on figure 10 are not consistent with the text.

   *We disagree with this point. A common critique of OSSEs is that we have no way of assessing the input uncertainties. Comparing simulations and observations is one such check so we think it is important support to the results. Posterior flux estimates will be the subject of an upcoming paper*

   *Number in the figure 10 has been corrected.*

2. The observation operator is different from several previous studies (e.g., Basu et al., 2013 cited in your paper). In equation (12), you interpret the averaging kernel to model levels. In a lot studies, the model vertical profiles are interpolated to the vertical levels of the retrievals, and pressure weighting function from retrievals is used in calculating model equivalent column $CO_2$. I think if the observation operator is done in this way,

you will not have the problem having to remove 1-second averaging observations if they span several grids.

This comment also was made by referee #1. We agree that this approach would solve that problem. However CMAQ data is at higher vertical resolution than the 20 levels of the OCO-2 retrieval. Running at least a simple interpolation from CMAQ to OCO-2 risks neglecting high resolution features in the CMAQ profiles. The averaging kernel is fairly smooth so the problem is less severe in this direction. It is a judgement call either way.

**4.4 Technical corrections**

1. Line 6 on page 11, seems missing a word.

   We have restructured the paragraph that starts in line 6 on page 11. We have added: "We solve the minimization with a change of variable $\vec{x}^b$. Given that our control vector $\vec{x}$ depends on the size of the multipliers of the principal eigenvectors of $\mathbf{B}$. Our vector $\vec{x}^b$ was reconstructed (as is given in Eq.11). This reconstruction includes a new vector $\vec{q}$, which is normalized the by the square-root of the eigenvalues of $\mathbf{B}$; this transformation involves minimization with respect to $\vec{q}$, rather than $\vec{x_p}$."

2. Line 3 on page 17, remove "uncertainty".

   Corrected

3. Line 3 on page 19, what could be other reasons? You used "partly" in the sentence..

   This comment also was made by referee #1. We have restructured the paragraph that starts in line 3 on page 19. We have added: "Another reason for a lower reduction in March compared to September is that in the northern region of Australia (the region where we assumed large uncertainties in March see Fig. 3a) winds come from primarily from the west (active monsoon). Prevailing winds in this zone restrict the ability of OCO-2 to constrain surface fluxes (primarily because we did not include OCO-2 soundings over the ocean). Taking into account only the number of soundings in September, we can see that the increase of the OCO-2 data (17%) has a significant impact on the percentage of uncertainty reduction of the prior flux."

4. Line 14 on page 25, double check the sentence. "the potential to more accurately observations"

   This comment also was made by referee #2. We have restructured the paragraph that start in line 14 on page 19. We have added: "Another important consideration in future work is that these flux inversions should be run with a finer temporal and horizontal resolution. Model simulations at higher temporal and spatial resolution are always in

better alignment with observation (fewer biases), mostly because they can sample closer to the measurement site location".

**References**

[revised manuscript text omitted]

---

## Author Response (AR2)

**Response to reviewers: ACP-2019-874**

**Yohanna Villalobos Cortés**

**April 19, 2020**

This document presents a point-by-point reply to the reviewer's comments on manuscript ACP-2019-874 (entitled 'The potential of OCO-2 data to reduce the uncertainties in $CO_2$ surface fluxes over Australia using a variational assimilation scheme'). This reply is written on behalf of all co-authors.

We would like to thank the reviewers for their comments and efforts towards improving our manuscript. The reviewer's comments are given in Roman type, and our replies are shown in blue.

**1 Response to referee #1**

**1.1 Technical corrections**

1. Page 3 line 1 put space after "version". Although Liang et al 2017 put version number as v02.xx and did not specify exact version, it is likely to be v02.21/02.31/02.41, collectively named as v02.2x.

   We have contacted the author of this article to clarify the version GOSAT data. Liang mentioned that version of the GOSAT is likely to be v2.60. Base on this information we have restructured the paragraph that starts in line 6 on page 11.

   Modified text: (FTS Level 2-3 data products; the product version is not given in the paper, but is likely to be version 02.60 (Liang, 2019)). We added this personal communication to the bibliography of the paper as well.

2. Page 3 line 3 add "of" after "precision

   Corrected

**2 Additional corrections**

We also have decided to modify labels of Figures 9, 10, 11 and 12. For example, label (a) Sensitivity case (1) in Fig. 9 was replaced by (a) Sensitivity case S1. This minor modification was made to be consistent with the text in the manuscript.

In section "Acknowledgements" we included the following text: "We acknowledge the contribution of the CMAQ adjoint team in providing us with the model code. We acknowledge the effort of Vanessa Haverd from CSIRO in providing us with the Australian biosphere carbon flux data".

This document includes a marked-up manuscript version (Latexdiff) showing these changes in the main report.

**References**

Liang, A. (2019). personal communication.

[revised manuscript text omitted]